# Stimulating at the right time to recover network states in a model of the cortico-basal ganglia-thalamic circuit

Timothy O. West[1,2], Peter J. Magill[1,3], Andrew Sharott[1], Vladimir Litvak[2], Simon F. Farmer[4,5], Hayriye Cagnan[1,2]*

1 Medical Research Council Brain Network Dynamics Unit, Nuffield Department of Clinical Neurosciences, University of Oxford, Oxford, United Kingdom, 2 Wellcome Centre for Human Neuroimaging, UCL Queen Square Institute of Neurology, London, United Kingdom, 3 Oxford Parkinson's Disease Centre, University of Oxford, Oxford, United Kingdom, 4 Department of Neurology, National Hospital for Neurology & Neurosurgery, London, United Kingdom, 5 Department of Clinical and Human Neuroscience, UCL Institute of Neurology, London, United Kingdom

* hayriye.cagnan@ndcn.ox.ac.uk

**Data Availability Statement:** The source code and data used to produce the results and analyses presented in this paper are available as a Zenodo

## Abstract

Synchronization of neural oscillations is thought to facilitate communication in the brain. Neurodegenerative pathologies such as Parkinson's disease (PD) can result in synaptic reorganization of the motor circuit, leading to altered neuronal dynamics and impaired neural communication. Treatments for PD aim to restore network function via pharmacological means such as dopamine replacement, or by suppressing pathological oscillations with deep brain stimulation. We tested the hypothesis that brain stimulation can operate beyond a simple "reversible lesion" effect to augment network communication. Specifically, we examined the modulation of beta band (14–30 Hz) activity, a known biomarker of motor deficits and potential control signal for stimulation in Parkinson's. To do this we setup a neural mass model of population activity within the cortico-basal ganglia-thalamic (CBGT) circuit with parameters that were constrained to yield spectral features comparable to those in experimental Parkinsonism. We modulated the connectivity of two major pathways known to be disrupted in PD and constructed statistical summaries of the spectra and functional connectivity of the resulting spontaneous activity. These were then used to assess the network-wide outcomes of closed-loop stimulation delivered to motor cortex and phase locked to subthalamic beta activity. Our results demonstrate that the spatial pattern of beta synchrony is dependent upon the strength of inputs to the STN. Precisely timed stimulation has the capacity to recover network states, with stimulation phase inducing activity with distinct spectral and spatial properties. These results provide a theoretical basis for the design of the next-generation brain stimulators that aim to restore neural communication in disease.

## Author summary

Diseases of the brain lead to a wide range of disabling symptoms for patients, by affecting their ability to move or think properly. These symptoms arise from disruption to both the

version controlled repository accessible at https://doi.org/10.5281/zenodo.5971846.

**Funding:** This work was supported by Medical Research Council UK Awards MR/R020418/1 (to H.C.), U138197109, MC_UU_12020/5, MC_UU_12024/2 and MC_UU_00003/5 (to P.J.M.), and MC_UU_12024/1 and MC_UU_00003/6 (to A.S.); Parkinson's UK Grant G-0806 (to P.J.M.); The Wellcome Centre for Human Neuroimaging is supported by core funding from the Wellcome 203147/Z/16/Z (V.L.). S.F.F. acknowledges funding support from the UCLH Biomedical Research Centre. The funders had no role in study design, data collection and analysis, decision to publish, or preparation of the manuscript.

**Competing interests:** The authors have declared that no competing interests exist.

organization of networks in the brain, but also the timing of neural activity that propagates around it. Treatments for disease with drugs can restore the organization of these networks to some extent, yet it is very difficult to deliver drugs with good spatial or temporal selectivity. Brain stimulation provides one way in which to improve the spatial specificity of treatment, yet understanding how to stimulate at the right time to achieve the best outcome for patients, remains an outstanding question. In this work we use simulations of an important circuit involved in Parkinson's disease that has parameters chosen to reflect recordings made in animal models of the disease. Using this computer model, we show how brain rhythms can act as signatures of underlying changes in networks. Further, we simulate intervention with temporally precise stimulation to show how future approaches to brain stimulation can act to restore or even augment neural networks following their degeneration in disease.

## Introduction

Current theories of communication in the brain hypothesise that phase synchronization [1] binds populations of neurons into transient assemblies [2] that facilitate the segregation and gating of information transfer [3]. Factors determining the efficacy of synchronous communication comprise: (A) the effective density of spikes received at the target- determined by the network of axonal tracts as well as the synaptic ultrastructure responsible for neural transmission, and (B) the receptivity of the target population determined by the temporal coordination of neuronal activity between the sender and receiver [4].

Neurodegeneration in diseases such as Parkinson's (PD) cause widespread changes in neural activity of the cortical-basal ganglia-thalamic (CBGT) circuit [5,6] leading to the motor impairments seen in patients. In terms of *factor A* (described above), degeneration of nigrostriatal dopaminergic neurons leads to the synaptic reorganization of the CBGT circuit [7]. Two important consequences are: i) the weakening of the cortico-subthalamic hyperdirect (HD) pathway [8–10]; and ii) the strengthening of the pallidal-subthalamic (PS) pathway [11,12]. Altered connectivity is hypothesized to disrupt temporal dynamics [13] (*factor B*; described above) such as the over-expression of beta frequency (β: 14–30 Hz) rhythms and synchrony, observed in patients [14] and animal models of disease [15]. Notably, these rhythms comprise transient intermittencies in amplitude (i.e., beta bursts) that become elongated and more frequent in Parkinsonism [16,17].

By modulating *factor A*, via synaptic reorganisation; or *factor B*, by altering the temporal relationship of activity between the sender and receiver, the efficacy of synchronous communication may be enhanced [4]. Importantly, either factor may be changed to compensate for a deficit in the other. Current pharmacological interventions in PD (e.g., Levodopa) principally target *factor A*- by restoring dopamine availability at the synapses. This has the secondary effect of altering neural activity (i.e. *factor B*) as evidenced by the reduction in beta power [18], attributable to a decrease in the rate and length of beta bursts [17]. Importantly, these changes may restore physiological transmission [19] yet pharmacological treatments are spatially non-specific and can induce side effects such as dyskinesias [20].

These two factors of neural communication can also provide targets for the improved design of therapeutic brain stimulation (e.g., deep brain stimulation- DBS). Conventional DBS reduces motor symptoms which are correlated with the suppression of beta band activity (i.e. *factor B*) [21] attributable to a shortening of beta bursts [22]. DBS and Levodopa both achieve similar therapeutic effects, but ostensibly in different ways: stimulation modulates neural

activity directly (particularly on the shorter time scale); whilst dopamine replacement causes changes in activity via the modulation of synaptic transmission. Nonetheless, current usage of DBS leads to effects similar to a surgical lesion [23]. Recent work suggests by refining the pattern of stimulation to target bursts of high amplitude beta activity [22,24] or altering its timing to match specific phases of the oscillation [25,26], it is possible to improve the specificity of therapy.

Unlike conventional DBS, phase specific stimulation can directly target oscillatory biomarkers of disease with the goal of improving neural communication [27–29] and attendant behavioural effects [30]. In Parkinsonism specifically, stimulation of the basal ganglia (such as the subthalamic nucleus STN, or internal globus pallidus GPi) has been shown to both amplify or suppress pathological beta oscillations, dependent on the specific phase targeted in both primate disease models [26] and patients [25]. Furthermore, the feasibility of multi-site control policies [31,32] has expanded the potential for phase specific stimulation to manipulate whole networks—with the possibility of delivering stimulation to one region of the brain (e.g., the STN) in closed-loop with that recorded at another (e.g., in the cortex).

We hypothesised that deficiencies in communication arising from synaptic reorganization in PD (i.e., *factor A*), may be compensated for by altering the timing of neural activity (i.e., *factor B*) through phase specific stimulation. To test this hypothesis, we used a neural mass model of the CBGT circuit constrained to spectral features of recordings made in a rodent model of PD. We summarise the changes in neural population activity (i.e., change to spectra and functional connectivity) associated with different network states (i.e., changes to structural connectivity) using statistical summaries that we term *spectral fingerprints* [33]. These summaries were estimated within transient bursts of activity that are significantly altered in PD [16,17]. We then used a model of on-line, phase-specific stimulation to establish how exogenous inputs can compensate for synaptic reorganization (i.e., factor A), by modulating the timing of neural activity (i.e., factor B) to selectively restore circuit wide patterns of synchronization. Specifically, following experimental work establishing dual-site stimulation and sensing [32,34], we explored both STN and motor cortex as potential sites for stimulation control and delivery. The results described here can inform the development of next-generation brain stimulation approaches that aim to recover and augment states of synchronous communication impaired in neurodegenerative disorders.

## Results

### Overview of results

We simulated a neural mass model of population activity propagating across the CBGT circuit to test four main hypotheses: (1) how can changes in "network state" (e.g. the strengthening or weakening of synaptic input to the STN) alter the spectral features of population activity; (2) on a wider scale, how do these same changes impact network wide phase synchronization; (3) how can phase specific stimulation compensate for synaptic changes, by mimicking spectral fingerprints associated with different network states; and finally, (4) how are the effects of stimulation dependent upon spontaneous alterations in the circuit's connectivity (e.g. due to further pathological or activity dependent synaptic reorganization). We frame our results in terms of Parkinsonian high amplitude beta oscillations that are known to be electrophysiological correlates of bradykinesia [14,35]. This gives focus to our central goal of informing the design of novel control algorithms to manipulate synchronous network activity associated with disease.

## A data constrained model of the cortico-basal ganglia-thalamic circuit exhibits synchronized beta band activity

We constrained the parameters of a neural mass model of the CBGT circuit to best fit the spectra and directed functional connectivity of data recorded from a 6-OHDA rodent model of PD [35]. A schematic of the model architecture, examples of time series and spectra from the recordings and simulations are presented in Fig 1. For the full set of data features (including the directed functional connectivity) and model fits, see supplementary S1 Fig. Properties of the nodes for which no empirical data were available (i.e., the GPi and Thal.), were inferred by the model fitting procedure (Fig 1B inset spectra; indicated with black solid lines).

Several qualitative features of the experimental data (reported in [36]) were well reproduced: 1) a beta peak was found across all spectra in the network (Fig 1B; inset spectra); 2) significant STN→M2 directed functional connectivity indicating feedback of beta oscillations

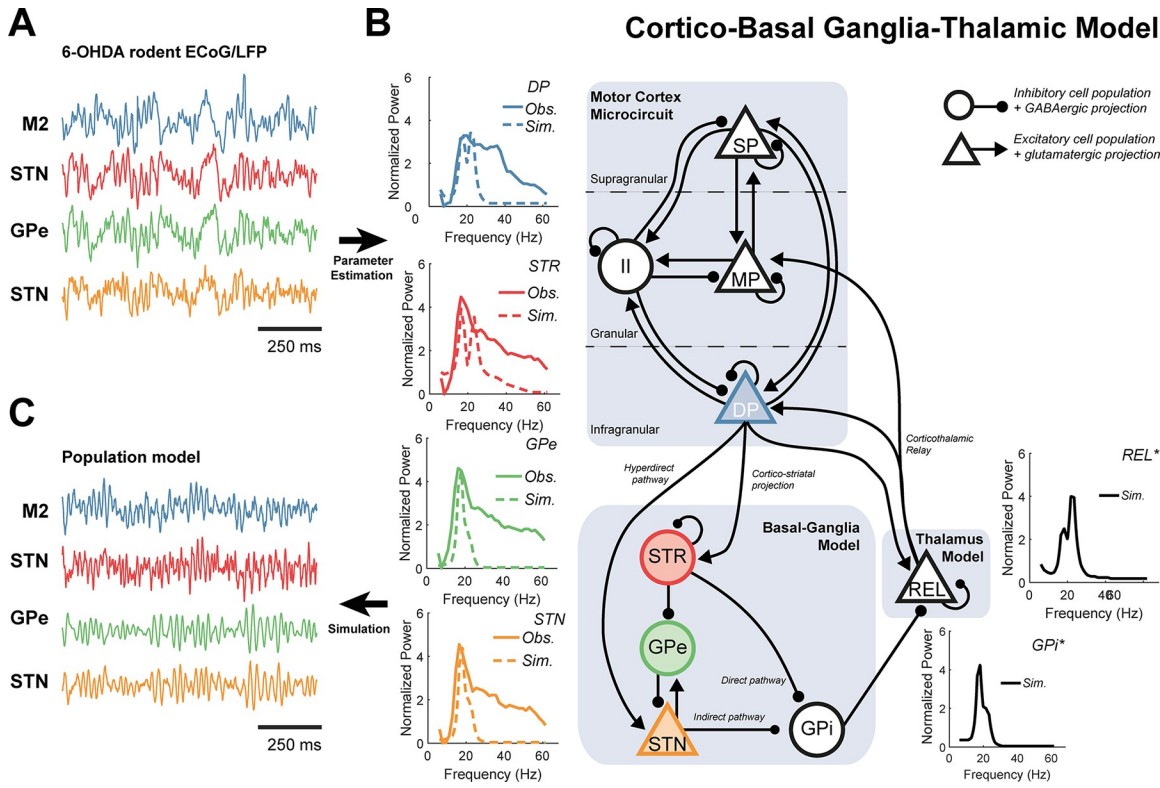

**Fig 1. Schematic of the cortico-basal ganglia-thalamic model and fit to empirical data from Parkinsonian rodents.** A model describing the population activity in this circuit was fit to data features (power spectra and directed functional connectivity) of **(A)** electrophysiological recordings: electrocorticography from motor cortex M2 (blue) as well as local field potentials from striatum (STR; red) external segment of the globus pallidus (GPe; green) and the subthalamic nucleus (STN; yellow) made in a 6-OHDA-lesioned rodent model of Parkinsonism. Data were normalized and band-passed at 4–100 Hz before being transformed to the data features used to estimate parameters. **(B)** Schematic of model architecture, detailing excitatory/glutamatergic projections (triangular nodes with arrows) and inhibitory/GABAergic projections (circular nodes with ball ended arrows). The motor cortex microcircuit comprises three layers: superficial pyramidal cells (SP; supragranular); middle pyramidal (MP; granular); and deep pyramidal cells (DP; infragranular), plus an inhibitory interneuron population (II). The basal ganglia model comprises four populations: STR, GPe, STN, and internal segment of the pallidus (GPi). The GPi forms the output of the basal ganglia and acts to inhibit relay cells of the ventrolateral thalamus (REL). GPi and REL were treated as hidden nodes and their respective neural activities were inferred from the dynamics of the empirically recorded brain regions. The main subcortical pathways include the direct, indirect, hyperdirect, and cortico-thalamic interactions. The inset graphs indicate the empirical and simulated power spectra in bold and dashed lines, respectively. For the full set of empirical and fitted data features please see S1 Fig. **(C)** Simulations of this circuit yields time series with transient, burst like behaviour similar to that seen *in vivo* (A).

from subcortex to cortex (supplementary S1 Fig); 3) stochastic "bursting" behaviour qualitatively similar to that in vivo (Fig 1A and 1C). Note that feature (2) is pronounced in the original 6-OHDA rodent data but has not been found in human data where functional connectivity is predominantly lead by cortical activity [37,38].

Nonetheless, model fits did not well reproduce the broadband high frequency activity (30–60 Hz) present in the data (Fig 1B; inset spectra). The model also produced activity at two sub-peaks at lower ($\beta_1$: 14–21 Hz) and upper ($\beta_2$: 21–30 Hz) beta frequencies, reflecting the bimodal structure of directed functional connectivity (see analysis of multiple Gaussian fits to features from individual animals; S2 Fig), and in line with experimental reports of a functionally related subdivision of the beta band [39,40].

## The strength of synaptic inputs to the STN from the pallidum and cortex leave distinct spectral features in STN/M2 population activity

We tested our first hypothesis that changes in connectivity leave their signature in the spectra of population activity. We examined how changes in subthalamic inputs from the HD and PS pathways modulate the spectra and coherence of the STN and M2, with the aim of using these spectral features to later classify the outcomes of cortical stimulation phase locked to $\beta_1$ rhythms sensed in the STN (see results section "Phase stimulation can Mimic the Spectral Features and Synchronization of Network Activity Relating to Altered Connectivity").

As introduced, experimental Parkinsonism results in *down-* or *up-* regulation of the hyperdirect (HD) and pallido-subthalamic (PS) pathways, respectively. In Fig 2, we show simulations in which the synaptic strengths of either the PS or HD pathways (panels A and B respectively) were altered to allow the modulation of spectral features between states. We defined an *Up-* and *Down-* state for both HD and PS pathways according to the properties outlined in the Materials and Methods section "Definition of Discrete Network Estates".

Since the parameters of the neural mass model were constrained using data from 6-OHDA lesioned rats, *PS-Up* (colour coded in figures with purple) and *HD-Down* (red) states reflect a hypothetical network associated with disease progression, and worsened symptom severity (Fig 2C). In contrast *PS-Down* (green) and *HD-Up* (blue) states reflect a dopamine intact network accompanying a reduction in disease symptoms and associated electrophysiological signatures of disease.

Simulations revealed that rhythmic activity in beta band was amplified further in the *PS-Up* and *HD-Down* states, whilst there was a marked reduction in STN and M2 beta rhythms in *PS-Down* and *HD-Up* states, as would be expected from the beta biomarker hypothesis of PD [14]. Fig 2D demonstrates that increasing the strength of the PS pathway (i.e., *PS-Up*) results in spectra with broadband amplification of beta ($\beta_1$ and $\beta_2$) power in the STN. Changes to cortical (M2) power spectra (Fig 2E) were more complex and exhibited a two-part response: (a) in the range of 30% to 100% connection strength, $\beta_1$ showed a higher sensitivity to connection strength changes than $\beta_2$, with the former increasing by ~80% from the *PS-Down* state up to that observed in the fitted model (i.e., the 6-OHDA lesioned state); (b) however, at connection strengths greater than 100%, a broadband amplification occurred resulting in spectra similar to that seen in the STN in *PS-Up* (described above). This is recapitulated by the response of the M2/STN coherence (Fig 2F) which follows changes in M2 power.

Examination of the spectra in the intermediate states linking *HD-Down* to *HD-Up* showed that strengthening this pathway above 160% exposes a transition in STN peak frequency from $\beta_1$ to $\beta_2$ (Fig 2G). This effect was also present in cortical (M2) activity (Fig 2H), but with the switch occurring at weaker connection strengths (~100% HD weight). Similarly, cortico-subthalamic coherence followed increases in M2 power (Fig 2I) with $\beta_2$ synchrony most

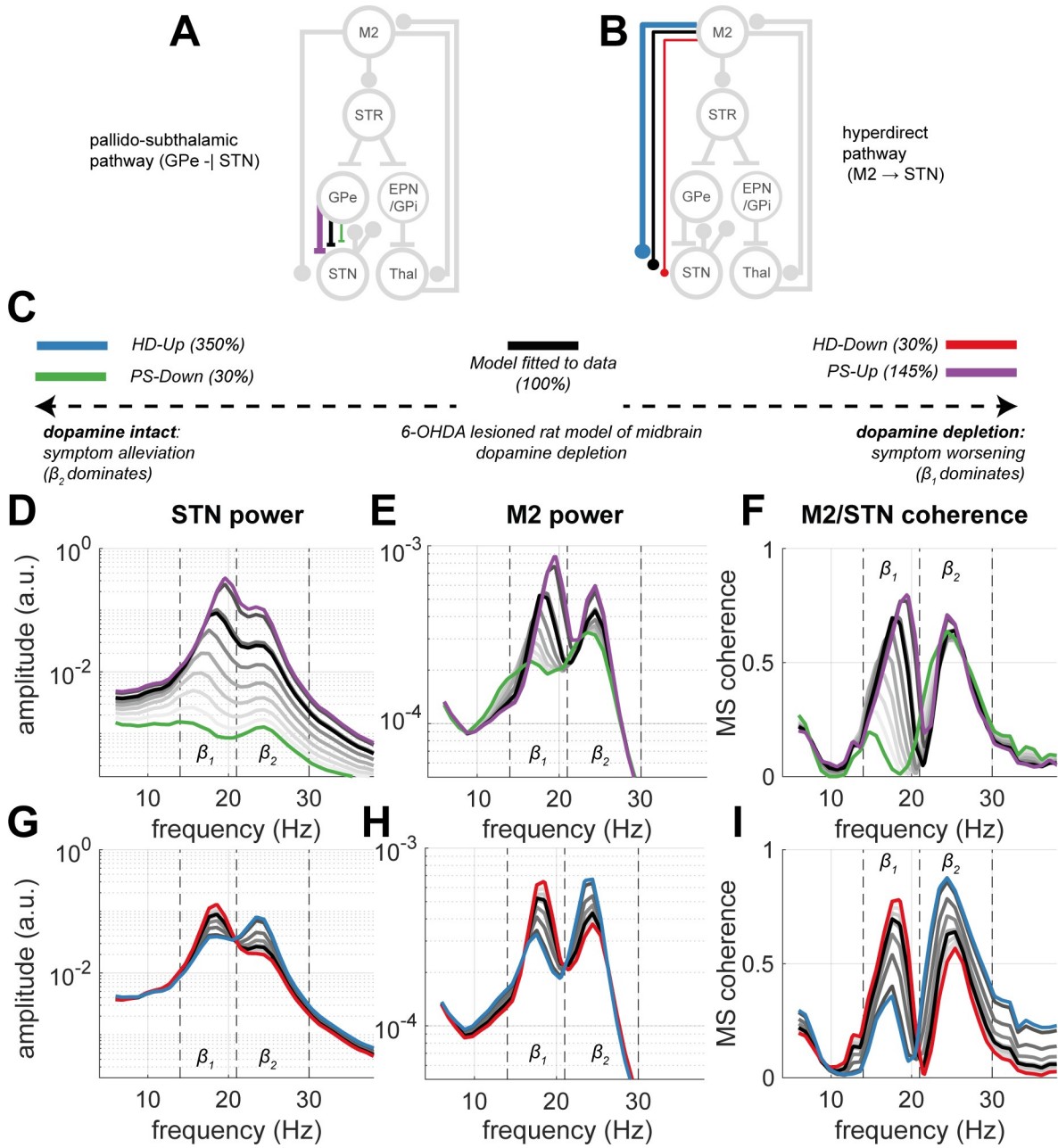

**Fig 2. Modulating the strength of inputs to the subthalamic nucleus (STN) from both hyperdirect (*HD*) and pallidosubthalamic (*PS*) pathways leaves distinct spectral features in the oscillatory activity and synchronization of cortex (M2) and STN.** Two discrete network states were defined for each pathway: a *Down*-regulated state fixed at 10% connectivity, as well as an *Up*-regulated state set at the connection strength eliciting ~200% increase in STN beta power. The fitted model (i.e., the 6-OHDA lesioned state, represented at 100% connectivity) is given in black. (A) Schematic of *PS-Up* (purple, 145% connection strength) and *PS-Down* (green, 30% connection strength). (B) Schematic of *HD-Down* (red, 30% connection strength) and *HD-Up* (blue, 350% connection strength). (C) Legend to the network states placed on a hypothetical scale from a dopamine intact state associated with a reduction in motor symptoms (far left), to the fitted model (with parameters constrained by data from 6-OHDA rat model of Parkinsonism; middle), to states indicating further progression of the pathology and worsening of motor symptoms (far right). (D) *PS* network states leave distinct spectral features in the power spectra of STN, with modulation occurring between lower ($\beta_1$) and upper beta ($\beta_2$) bands. (E) Similar responses can be seen in the M2 power spectra; as well as in (F) the functional connectivity between STN and M2 in terms of the magnitude squared coherence. (G, H, and I) Same as (D, E, and F) but for the *HD* defined states. Grey lines show the intermediate spectra generated between Up and Down states.

predominant in the *HD-Up* state. A similar differential response of beta sub-bands in Parkinsonism has been reported experimentally [40] and will be used as the main spectral discriminator between *HD-Down* and *HD-Up* states in this model.

## The strength of hyperdirect and pallidal inputs to STN shape network-wide patterns of phase synchronization

We next tested our second hypothesis that changes in connectivity (i.e., the strength of PS or HD inputs to the STN) can impact synchronization across the entire CBGT circuit. To this end, we investigated the phase synchronization occurring within bursts of rhythmic activity since these are significantly altered in Parkinsonism [16,41,42], and are commonly targeted with closed-loop stimulation (results section "A Model of Dual-site Controlled Phase Locked Stimulation can Effectively Modulate Spectral Features of Population Activity"). Bursts were defined by setting a threshold on the envelope of STN activity band-passed at beta frequencies (see Materials and Methods section "Definition of Transient Burst Events and Statistics" and Fig 3A). Due to the bimodal nature of the STN spectra, bursts, and phase synchronization (within-burst phase locking value- PLV) were derived separately for $\beta_1$ and $\beta_2$ frequencies.

We used simulations from the four previously defined *Up* and *Down* network states to construct connectivity matrices (Fig 3B) representing the change in the within-burst phase locking value (PLV) from that measured in simulations of the fitted model (significance tested against surrogate distribution–see Materials and Methods section "Phase Synchronization: Connectivity Matrices and Time Resolved Estimates", permutation-test (500), $\alpha^* < 0.05$). In the two states associated with an amelioration in motor symptoms (*PS-Down* and *HD-Up*), we found two major effects: (a) downregulation of the PS connection to 30% of its strength in the fitted model (i.e., in the dopamine depleted state) significantly reduced cortical and thalamic $\beta_1$ phase synchrony (*PS-Down;* Fig 3B, groups labelled I and II). (b) The switch to $\beta_2$ frequencies in the STN and M2 induced by strengthening of the HD pathway (described in the previous section) is also seen in increased cortical and thalamic $\beta_2$ synchrony with the rest of the network (groups III and IV respectively), and occurs simultaneously to changes in $\beta_1$ synchrony in the same regions (groups V and VI).

When looking at the states associated with disease progression, we found that strengthening pallidal inhibition (i.e., *PS-Up*) resulted in a selective desynchronization between STR and its downstream targets in the indirect pathway (GPe, STN, and GPi) at $\beta_2$ frequencies (group VII). This suggest that increased pallidal inhibition of the STN results in the loss of frequency selective synchronization of the indirect pathway. Furthermore, downregulation of the HD pathway (*HD-Down*) was associated with significant decreases (from the fitted model) in $\beta_2$ cortico-subthalamic PLV (group IX). We refer to the switching of synchronization between $\beta_1$ and $\beta_2$ frequency associated with modulation of the HD pathway strength as a "conditioning" effect, that we will later relate to the effects of cortical stimulation.

## Examination of the phase reorganization of population activities during transient beta bursts

To understand how transiently phase synchronized networks emerge during bursts of high amplitude beta activity and how they are shaped by synaptic inputs to the STN, we next performed a set of time resolved analyses of the phase alignment between M2 and STN regions (Fig 3C–3J). Since STN rhythms in $\beta_1$ were more prominent than $\beta_2$ in the model fitted to the 6-OHDA lesioned state, and in states associated with worsening of symptoms (i.e., *HD-Down*, shown in Fig 2), we used this sub-band to define the bursts and estimate changes in the dynamics of phase synchronization during beta bursts. We specifically looked at transient

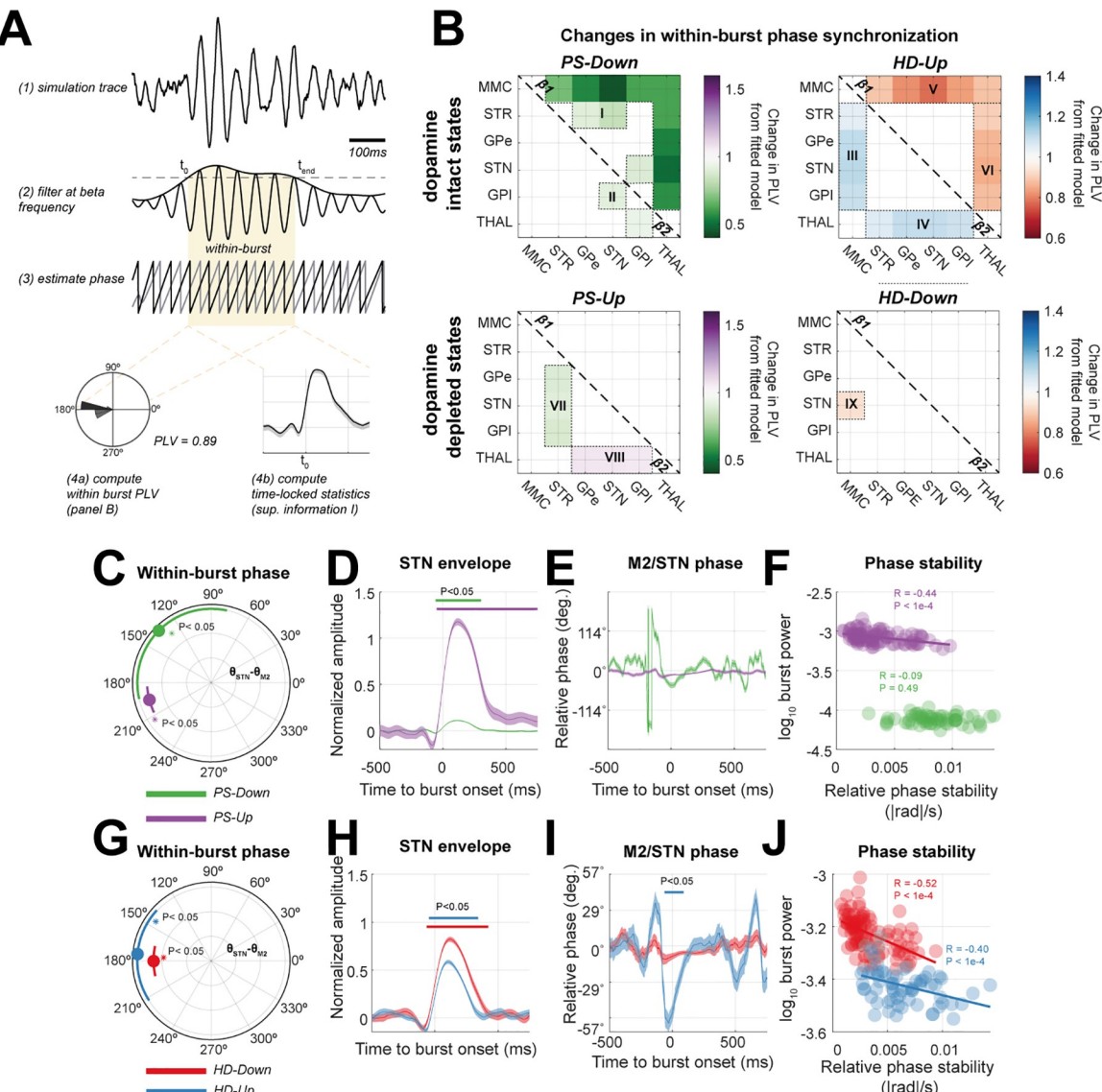

**Fig 3. Changes in hyperdirect (*HD*) or pallido-subthalamic (*PS*) network states result in circuit wide alterations in transient (i.e., within-burst) phase synchronization at beta band frequencies.** Data were simulated using the four predefined network states (see main text and Fig 2) of the CBGT model: *PS-Up* (purple), *PS-Down* (green), *HD-Up* (blue), and *HD-Down* (red). (A) 120s of data were simulated from each model and then bandpass filtered at lower or upper beta band. The Hilbert envelope was used to defined "burst" epochs as periods of suprathreshold (>75th percentile) activity in the STN (shaded yellow area). Phases across all six sources in the model were reconstructed using the angle of the Hilbert transformed signals. This phase estimate was used to construct connectivity matrices or time-locked statistics of burst activity. (B) Connectivity matrices indicating all possible pairs of within-burst phase synchronization (PLV) across the CBGT circuit. Matrices show the difference in PLV from that of the fitted model, color-coded according to the inset colorbar. The matrix is thresholded to only show significant changes in PLV from those estimated in the fitted model (compared to surrogate distribution, permutation-test (500), $\alpha < 0.05$). Results for lower and upper beta are shown in the top and bottom diagonals respectively. (C and G) Radar plot of within-burst (i.e., when the STN envelope is suprathreshold) changes in STN/M2 phase difference. Circles indicate median, with bars giving the circular standard deviation. *indicate significant Rayleigh test for difference in mean phase from those computed from length matched, randomly selected out-of-burst data. Note that the radial dimension has no meaning, bars are offset for presentation purposes. Angular lengths should be interpreted with relation to the gridlines. (D and H) Analysis of the STN amplitude envelope between *PS-Up/Down* and *HD-Up/Down* filtered at lower beta frequencies. Traces are mean +/- S.E.M timelocked to burst onset at t = 0. Bars indicate significant cluster-statistics for deviation from length-matched, out-of-burst data (cluster-statistic (500), $\alpha < 0.05$). (E and I) Analysis of M2/STN phase difference (centred relative to the mean phase at 0˚). (F and J) Scatter plots of burst amplitude versus the relative phase stability (estimated as the mean absolute derivative of STN/M2 phase difference in the window 0 to +500 ms). In the case where there was a significant Pearson's correlation coefficient (R), we plot a regression line. Overall burst amplitude was correlated with phase stability.

dynamics in phase corresponding to $\beta_1$ bursts, as these periods of activity are later targeted with phase specific stimulation (results section "A Model of Dual-site Controlled Phase Locked Stimulation can Effectively Modulate Spectral Features of Population Activity").

Population activities from the up or down modulated network states exhibited a reorganization in their within-burst phase alignment relative to that observed in the fitted model (Fig 3C and 3G; first column). The *PS-Up* state exhibited a close to anti-phase relationship (203±5˚) between M2 and STN and was found to be similar to that for *HD-Down-* the other state associated with worsened motor symptoms (panel H; red line, 172±29˚). Interestingly, both *HD-Up* and *PS-Down-* states associated with weakened $\beta_1$ amplitude (and accordant reduction in PD symptoms)- exhibited a much larger variance, with that of *PS-Down* approaching a 45˚ circular standard deviation. This suggests that an optimal phase alignment exists for promotion of $\beta_1$ bursts, a feature that we attempt to leverage when implementing phase specific stimulation.

Across all simulated states, there was a rapid adjustment of cortico-subthalamic phase difference around burst onset (panels E and I), as seen by the initial transient phase reorganization that is most prominent prior to peak amplitude within a burst. However, the phase evolution in the states associated with a reduction in Parkinsonian symptoms and weaker $\beta_1$ burst amplitudes (*PS-Down-* green trace, or *HD-Up*–blue trace) exhibited: (a) a larger transient deviation in phase, and (b) noisier dynamics that settled more slowly over the duration of the burst after the burst amplitude started reducing. Note that the direction of change in relative phase was not consistent across different states.

Corroborating this, analysis of the stability (in terms of the mean rate of change) of the phase difference between STN and M2 activity during a $\beta_1$ burst (i.e. at 0 to 750ms after burst onset) using the Relative Phase Stability metric (see Materials and Methods section "Phase Synchronization: Connectivity Matrices and Time Resolved Estimates") directly correlated with STN $\beta_1$ burst power in three out of the four states (Fig 3F and 3J; Pearson's R, P $\leq$ 0.001). This supports the idea that maintenance of high amplitude activity accompanies periods of stable phase locking between STN and M2, although it does not determine whether these changes in phase drive those in amplitude or vice-versa (see results section "A Model of Dual-Site Controlled Phase Locked Stimulation Can Effectively Modulate Spectral Features of Population Activity" below for a direct manipulation of phase in the model).

## A model of dual-site controlled phase locked stimulation can effectively modulate spectral features of population activity

We next examined our third hypothesis that phase-specific stimulation could modulate oscillatory activity using a model of on-line, closed-loop stimulation with a phase-locked control algorithm. We devised a dual-site control policy using STN as the sensing site (to gate and parameterize stimulation) and M2 as the stimulation site (cartooned in Fig 4A). Since STN rhythms in $\beta_1$ were the most amplified features in states associated with worsening of symptoms (i.e., *HD-Down* and *PS-Up*) we targeted this band with closed loop stimulation. Cortical stimulation was delivered at the onset of a $\beta_1$ burst sensed in the STN and was locked to different phases of this activity. This stimulation/sensing pair was chosen as it provided superior STN beta suppression than the alternative scenario with STN stimulation with cortical sensing for the same stimulation thresholds and amplitudes (supplementary S4 Fig). Furthermore, supplementary analyses (S7 Fig) showed that flexible state recovery (a principle goal of this work; see results section "Phase stimulation can mimic the spectral features and synchronization of network activity relating to altered connectivity") with STN stimulation was inferior when compared to the cortical stimulation presented. Stimulation was modelled through the addition of a sinusoidal voltage to the average membrane potential of the superficial cell layer

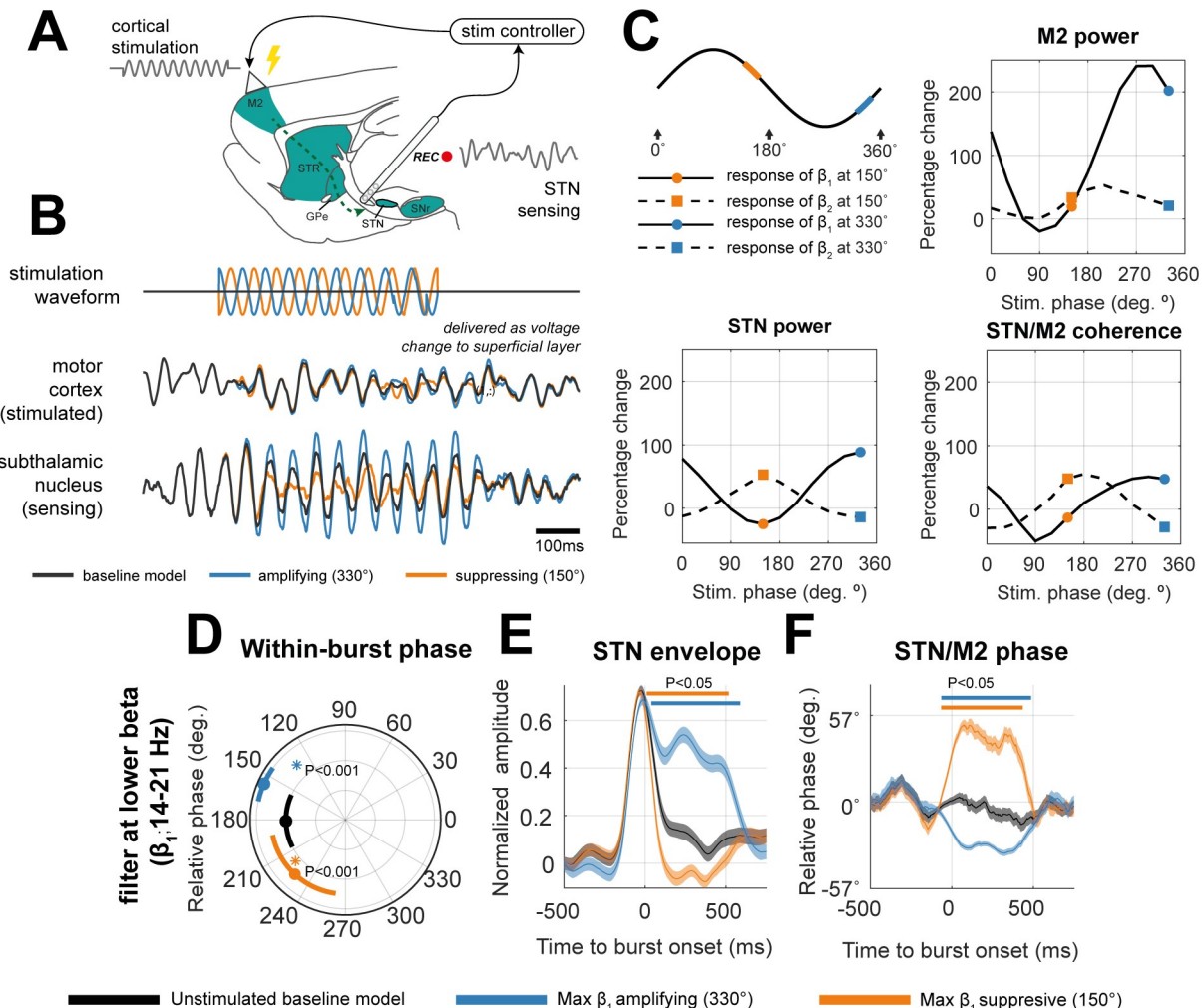

**Fig 4. A model of dual-site controlled motor cortical stimulation phase locked to signals sensed in the subthalamic nucleus, can modulate beta activity across the network. (A) Schematic for phase specific neuromodulation in the rodent brain.** Activity at the STN is used to gate stimulation delivery at M2. **(B)** Stimulation waveforms were generated using an 18 Hz sinusoid, with phase constructed to be offset against an on-line phase estimate of STN activity. Stimulation was delivered as a direct voltage injection to the stimulated population (superficial layer of M2). Effects upon β power were either amplifying (blue; 330˚) or suppressing (orange; 150˚) depending on the phase. **(C)** Amplitude response curves of power in the motor cortex, STN, as well as their synchronization (in terms of coherence) when sweeping across 12 stimulation phases. Curves are shown separately for lower ($\beta_1$; bold line) or upper beta ($\beta_2$; dashed line). Corresponding spectra are shown in supplementary S5 Fig. **(D)** Analysis of mean M2/STN phase difference during stimulation period (centred relative to the mean phase at 0˚) filtered at lower beta frequencies. Circular markers indicate median, with bars giving the circular standard deviation. *indicate significant Rayleigh test for difference in phases from the unstimulated, fitted model. Note that the radial dimension has no meaning, bars are offset for presentation purposes. Angular lengths should be interpreted with relation to the grid-lines. **(E)** Analysis of the STN amplitude envelope during stimulation. **(F)** Analysis of M2/STN phase difference tracked across time. Traces are mean+/- S.E.M time locked to burst onset at t = 0. Bars indicate significant cluster-statistics for deviation from the unstimulated model (two-sample t-test, n = 500, $\alpha < 0.05$).

in the cortical microcircuit. Input amplitude was fixed to 1/3 of the intrinsic noise level and the central frequency was set to 18 Hz. The phase was constructed to preserve a fixed phase alignment with the bandlimited signal (at $\beta_1$) sensed from the STN population. We also tested a number of non-phase specific control strategies, including sinusoidal 18 Hz (with random phase), as well as playback of phase-locked stimulation (see Materials and Methods section "Modelling Phase Locked Stimulation of Motor Cortex Using Activity in the Subthalamic Nucleus"). None of these alternative stimulation strategies were able to achieve a suppression of beta rhythms in the model (supplementary S3 Fig; see also discussion).

An example trace activity observed during stimulation is given in Fig 4B. This shows: (a) how the controller tracks phase and delivers a stimulus (to M2) with a phase shift relative to the sensed population (STN). Phase estimation is impaired towards the end of suppressing stimulation (e.g., orange traces) due to the reduced SNR; and (b) the effects of stimulation upon the rhythmic activity in the STN are dependent upon the specific phase, with some amplifying and others suppressing (examples in blue and orange respectively). The stimulation angle with respect to the underlying sine wave is shown in the first panel of Fig 4C. The effects upon the power, and coherence between the motor cortex and STN are summarised as amplitude response curves (ARCs) in the remaining panels of Fig 4C. The total range of modulation for STN $\beta_1$ (bold lines) is -18% to +155%, and for $\beta_2$ (dashed line) is +10% to +60% compared to the power in the unstimulated model. The total range of modulation for M2 power $\beta_1$ is -17% to +240%, and for $\beta_2$ is 0% to +40%.

## The suppressive and amplifying effects of phase locked stimulation can be explained in terms of their effects upon phase progression within bursts

To understand how the effects of stimulation upon transient burst activity compare to that occurring spontaneously (i.e., the analyses in Fig 3).–we investigated phase coupling in the $\beta_1$ band (i.e., the frequency of activity sensed in the STN and targeted by stimulation) between the STN and motor cortex (i.e., the sensing and stimulating sites respectively) during phase specific cortical stimulation. Fig 4D shows that cortical stimulation induces a significant shift in the phase difference between STN and M2 for the duration of stimulation, for both amplifying (blue; 151±10˚; compared to the unstimulated model; Watson-Williams (103) = 13.45; P < 0.001) or suppressing (orange; 221±11˚; compared to the unstimulated model; Watson-Williams (103) = 31.66; P < 0.001) stimulation. Notably, the phase was shifted in opposite directions corresponding to whether stimulation was $\beta_1$ amplifying (closer to in-phase) or suppressing (closer to anti-phase).

Analysis of the burst envelopes (panel E; clusters indicate significant shift from unstimulated model; indicated by bold bar above traces) shows that amplifying effects of stimulation (blue; targeting 330˚) are achieved by sustaining and moderately amplifying $\beta_1$ amplitude across the burst, whilst suppressive effects (orange; targeting 150˚) arise due to a rapid shortening of the burst duration. Accompanying changes in cortico-subthalamic phase alignment during stimulation (panel F) were different to that observed spontaneously (i.e., in Fig 3). Previously, changes in phase around the burst were predominantly made up of a transient slip near the initiation of the burst. Instead, when stimulation was applied, there was a phase realignment that is sustained across the stimulation period.

Phase locked stimulation does not always induce a change in phase alignment, as we demonstrate when the same analysis was performed at $\beta_2$ frequencies (i.e., the non-targeted frequency, supplementary S5 Fig). Changes in within-burst $\beta_2$ dynamics are delayed (approximately +100 ms after stimulation) suggesting that the emergence of these rhythms in the STN potentially result from the rapid suppression of $\beta_1$ rhythms as was shown in Fig 4C, and could result from propagation of a cortically derived $\beta_2$ rhythm.

## Phase stimulation can mimic the spectral features and synchronization of network activity relating to altered connectivity

We then tested the hypothesis that the phase specificity of stimulation can provide selective modulation of the spectra and network-wide changes in synchronization of rhythmic activity that can mimic changes in synaptic connectivity. To do this we compared the spectral fingerprints of spontaneous population activity following alterations of synaptic connectivity

(depicted in Figs 2 and 3B), with those estimated from activity during stimulation (depicted in Fig 4). These summaries of population activity (Fig 5A–5C) were computed using two statistical features: (a) the concatenated power spectra of the six regions in the model; and (b) the connectivity matrix indicating the magnitude of the pairwise complex PLV (see Materials and Methods section "Characterizing Network States with Spectral Fingerprints"). Fingerprints between stimulated and spontaneous data were then compared for similarity using the pooled $R^2$. The results of this analysis are presented in Fig 5D and 5F and show that depending upon the feature used to summarise the circuit activity, different phases of stimulation can trigger activity that resembles to different degrees (in terms of the pooled $R^2$) each of the four previously defined network states.

In Fig 5D, comparison of the local power spectra between stimulated and spontaneous states shows that, cortical stimulation phase-locked to STN $\beta_1$ activity can evoke rhythmic activity resembling either the *HD-Up/Down* or *PS-Up* states (>60% of the explained variance). Notably there was an anti-phase relationship between stimulation outcomes with spectral fingerprints most resembling the *HD-Down* (peak $R^2 = 0.96$ at 330˚) and the *HD-Up* states (peak $R^2 = 0.72$ at 150˚). This antiphase recovery of states mirrors that of the switch between $\beta_1$ and $\beta_2$ power in the STN (shown in Fig 4C), but goes beyond this to show that cortical stimulation modulates spectral features across the wider network. This argument is reinforced by comparisons between stimulation outcomes and patterns of network-wide phase synchronization (Fig 5E). Recovered states most resembled *Up-* or *Down-* regulation of the HD pathway, but also showed differences with the mapping using just power spectra. In particular there was a peak in the recovery of *PS-Down* (peak $R^2 = 0.76$ at 90˚) when the recovery of *PS-Up* was at its lowest (trough $R^2 = 0.56$ at 90˚). The difference in recovery with summaries of either spectral or phase synchronization suggests a disconnect between the two features, with stimulation more readily changing interregional synchronization than local power spectra. Furthermore, the ability to accurately recover states related to an upregulated HD pathway, suggests that M2 stimulation phase locked to $\beta_1$ activity in the STN may have the potential to restore network activity following degeneration of this same pathway in response to dopamine depletion.

In line with this, recovery of network states, linked to empirically derived control/lesion connectivity patterns (supplementary S6 Fig), exhibited an antiphase relationship. Note however, the ability of stimulation to recover spectral fingerprints derived from dopamine intact control animals ($R^2 < -1$) was poor. Spectral profiles derived from control animals did not contain a distinct peak in the beta band which gave rise to this poor performance. Complete suppression of beta rhythms is not reproducible with the ~17% maximal $\beta_1$ suppression achievable with modelled stimulation. These data reflect not only hypothesised changes in HD and PS pathways, but reconfigurations of the wider network expected to accompany dopamine depletion.

When both spectral and phase synchronization features were combined to give an overall mapping of stimulation phase to states (Fig 5D), we confirmed the ability of phasic stimulation to broadly mimic activity matching *Up-* or *Down-* regulation of the HD pathway, as well as capture > 60% of the variance of features associated with the *PS-Up* state. None of the stimulation phases achieved over 45% explained variance of features derived from the *PS-Down* state.

## Phase locked stimulation can compensate for changes in synaptic connectivity by mimicking network states

Finally, we test to what extent the recoverability of states is limited by the network connectivity. We formed a set of secondary models in which the strength of either the HD or PS pathways was modulated across a continuous scale, and then examined the match with fingerprints

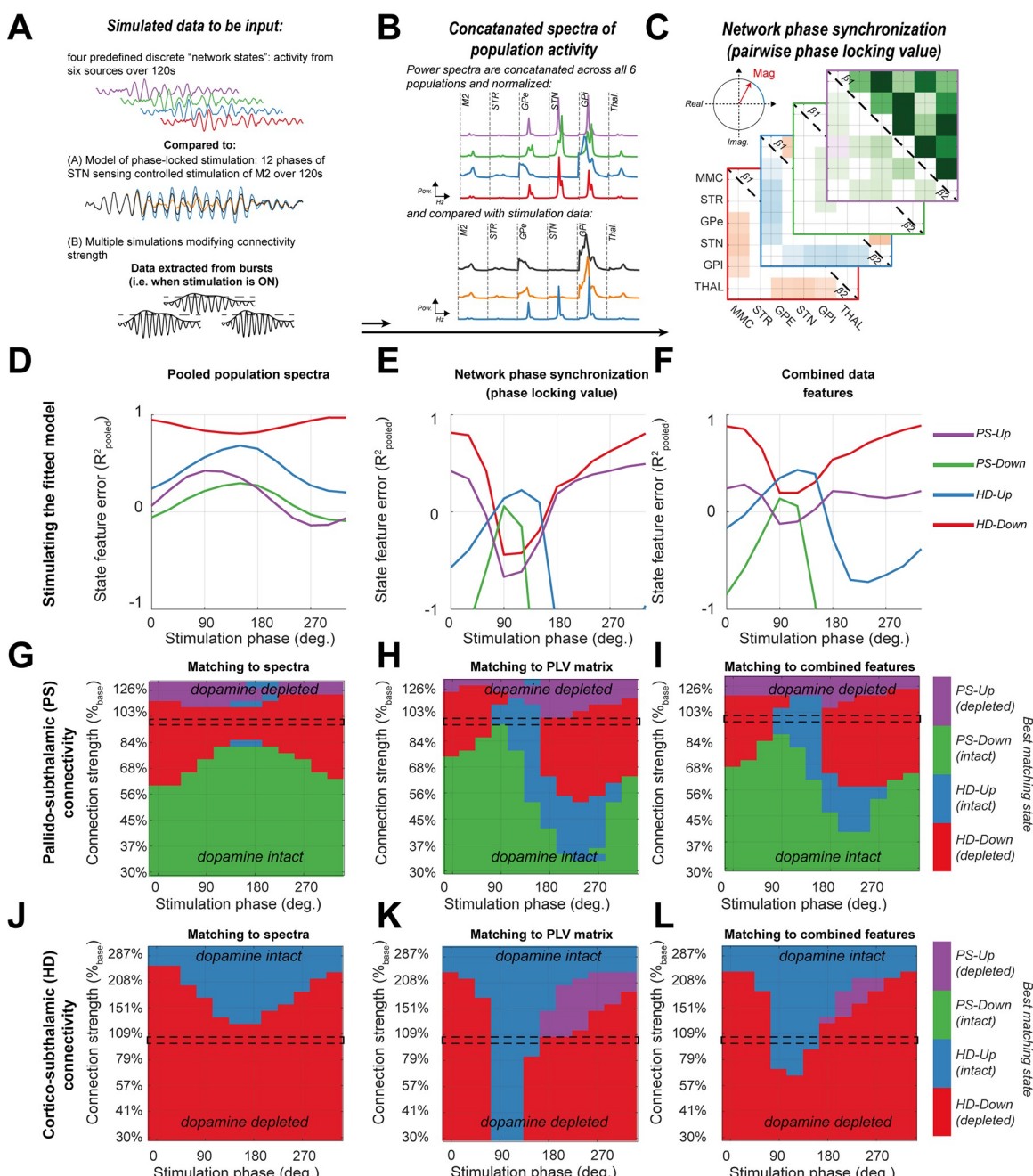

**Fig 5. Stimulation of the cortex phase locked to activity sensed in the STN can induce network-wide patterns of activity that resembles those corresponding to spectral fingerprints of spontaneous activity associated with changes in connection strength.** (A) Data simulated from the four network states *PS-Up* (purple), *PS-Down* (green), *HD-Up* (blue), and *HD-Down* (red) (see main text and Fig 2) was used to compute spectral fingerprints of spontaneous burst activity which were then compared against those constructed from the outcomes of phase locked stimulation. Only data during which stimulation was delivered (i.e., within a high amplitude STN beta burst event) was analysed. We used two different spectral fingerprints to compare network states with stimulated epochs: (B) the concatenated power spectra of each of the 6 populations in the CBGT circuit. Data were normalized to unit variance prior to concatenation such that comparisons were focused on the shape of spectra rather than large differences in amplitude. (C) Matrices of the pairwise phase synchronization (magnitude of the PLV) were estimated using the Hilbert transformed signals (see Fig 3A) for both $\beta_1$ and $\beta_2$ frequencies. (D, E, and F) Stimulation was applied across 12 phases in the fitted model and data were compared (using pooled $R^2$) to each network state using: the concatenated spectra (D); the matrix of PLV magnitudes (E); and the two features combined (F). These results show that spectral fingerprints from stimulation outcomes resembles that from different network states (i.e., a change in synaptic connectivity) depending upon the phase at which stimulation was delivered. (G, H, I) The above was repeated but when varying the connection strength of the PS pathway. Results are plot as a heatmap color-coded to indicate the best fitting state at each phase (x-axis;

angle of stimulation relative to STN activity) and connection strength (y-axis; percentage of fitted synaptic strength, dashed line indicates 100%- i.e., the model fit to the 6-OHDA lesion data and plot in D, E, and F). (J, K, and L) Same as G, H, and I, but for modulations of HD pathway strength.

of spontaneous activity in the four discrete network states used throughout the paper (Fig 5G–5I and 5J–5L respectively). Results show that the states accessible are dictated by the strength of inputs to the STN. At weakened PS pathway strengths (close to ~80%,) stimulation is most flexible, and able to access 3 out of 4 states (Fig 5G), with the range increasing when using phase synchronization as the feature to be compared (Fig 5H). As pallidal inhibition of the STN is weakened, it becomes harder to mimic the patterns of activity linked to the *PS-Up* state as subcortical synchronization at $\beta_1$ frequencies is effectively arrested.

Similarly, both strengthening or weakening of the HD pathway reduces the range of network states accessible with stimulation (Fig 5J–5L). Again, fingerprints of phase synchronization related to *HD-Up* or *HD-Down* were most readily accessed, with both recoverable when HD strength was in the range 70–311% (Fig 5K). These results suggest that the capacity of well-timed stimulation to recover both local and network signatures will be moderated by the strengths of connectivity within the CBGT circuit.

## Discussion

### Summary of findings

Using simulations of the CBGT circuit (Fig 1), we have shown that network-wide synchronization, disrupted by altered synaptic connectivity, may be restored by well-timed stimulation. In agreement with our first two hypotheses, the PS and HD pathways, both significantly altered in Parkinsonism, can determine the expression of beta frequency biomarkers, leaving distinct fingerprints in both the power spectra (Fig 2) and patterns of network-wide phase synchronization (Fig 3). Notably, the HD pathway played a conditioning role by dictating the relative expression of $\beta_1$ (14–21 Hz) or $\beta_2$ (21–30 Hz) activity, whilst the PS pathway acted as a broadband modulator of beta amplitude. To test our third hypothesis, we showed that M2 stimulation phase locked to STN can achieve both focal effects- through amplification or suppression of rhythmic activity (Fig 4), and global effects- by altering circuit-wide synchronization. In support of this hypothesis, stimulation yielded spectral fingerprints that well matched states defined under altered connectivity (Fig 5D and 5F). Finally, we showed that the ability of stimulation to recover network states is dependent upon the organization of the network at the time when an input was given (Fig 5G–5L), with the HD states more readily recovered than PS states. These results support the idea that phase locked stimulation can restore or compensate for deficits in synchronous communication arising from large scale synaptic reorganization in diseases such as Parkinsonism.

### Different network states underlie the expression of oscillatory control signals and determine the effects of stimulation

Neural oscillations are important biomarkers for a range of neuropsychiatric disorders [43]. These spectral features, which often reflect symptom severity, can also be used as control signals in closed-loop brain stimulation [42]. In Parkinsonism, STN beta activity has been used to control closed-loop DBS [42], where its effective suppression leads to a concurrent reduction in motor symptoms [44,45].

Our model shows that the expression of STN beta is dependent on network connectivity with the PS pathway controlling the gain of a subcortical STN/GPe resonator, that increases

the amplitude of broadband beta activity STN, but at the expense of spatial and frequency selectivity of network synchrony (Fig 3B). This agrees with studies implicating increased coupling of the STN/GPe loop in the emergence of excess beta synchrony in the CBGT circuit [46,47], and with experimental evidence that this pathway is strengthened [11,12] and overactive [48] following dopamine depletion.

Furthermore, we found that the HD pathway shifts beta rhythms to higher frequencies in line with the hypothesis that $\beta_2$ activity is a signature of cortico-subthalamic drive, following the finding of a correlation between $\beta_2$ power and HD tract density in PD patients [40]. Whether $\beta_1$ or $\beta_2$ bands reflect functionally distinct rhythms is not well understood- lower frequency activity is more readily suppressed by Levodopa [18] and DBS [38], and STN activity in the two sub-bands synchronize with different cortical regions [38,39]. In agreement with the model presented here, recent experimental evidence suggests that the specific phase and frequency of beta expression is modulated by changes in dopaminergic tone [49]. These results highlight how singular changes to monosynaptic inputs to the STN can shape activity of the wider network, not just by altering the amplitude and frequency of oscillatory activity, but also by dictating the synchronization of different elements of the wider circuit.

Taken together, the results presented agree with several computational [13,50] and experimental models [51] that show that pathological STN synchrony arises from the interplay of PS and HD pathway inputs impinging on the STN. Weakening of the HD pathway may give rise to STN/GPe resonance at β1 frequencies, as resonant activity is undisrupted by mismatched β2 inputs arriving from the cortex. In agreement with other models exploring thalamocortical feedback [13], we suggest that β1 rhythms amplified by the STN/GPe loop can return to motor cortex- effectively reinforcing synchrony across the CBGT at this frequency. This potentially explains how reduced HD transmission in PD can be observed alongside elevated cortico-subthalamic coherence [9]. How this translates to enhanced STN firing rate seen in primate models [52] is unclear, and would require models incorporating both oscillatory dynamics and detailed synaptic conductances to explain. Our model is distinct from a previous dynamic causal modelling study in which effective connectivity of the HD pathway was found to be raised in the 6-OHDA state [50], a finding in disagreement with findings of impaired cortico-subthalamic efficacy [8,9], and likely arising due to the relative insignificance of the STN/GPe circuit in that model.

## Phase locked stimulation provides a means by which to target and manipulate synchronous interareal communication in the brain

Network perspectives of brain stimulation [53] enables therapies to go beyond the focal targeting of rhythms, and towards modulation of functional networks such as those disturbed in neuropsychiatric disorders spanning depression, anxiety, and dementia [54,55]. Here, we show that stimulation of the motor cortex not only manipulates beta rhythms in the STN, but also repatterns the phase synchronization of the circuit in a way that could bias communication [1]. We report here that stimulation of M2 could achieve a maximum of ~30% suppression of STN $\beta_1$ rhythms, similar to the ~25% suppression achieved experimentally in a 6-OHDA rodent model [31]. Similar suppressive effects were also observed during adaptive DBS in humans (~14% beta power reduction in [44]) which was linked to a ~50% reduction in blinded hemi body UPDRS. It should be noted that approximately 60% suppression of beta power is observed following levodopa administration [56]. The range of amplification and suppression reported here is dependent upon both the threshold level chosen to gate stimulation delivery and stimulation amplitude (see supplementary S4 Fig). We also found that stimulation more readily amplified than suppressed the target rhythm, an asymmetry likely arising from:

(A) the maximum limits of suppression (-100%) vs amplification (unbounded); (B) the relative ease with which it is possible to entrain the system to an external input delivered in open loop, than it is to use a precisely timed perturbation to completely suppress an oscillation.

Phase locked stimulation in our model can selectively pattern network activity in a way that most readily resembles up- or down-regulation of the cortico-subthalamic HD pathway. Whilst the preference for stimulation to recover network synchrony related to the HD states (Fig 5) may arise due to our choice of sensing and stimulation sites (see discussion below), the coexistence of $\beta_1$ and $\beta_2$ frequency activity in this model means that targeted suppression of $\beta_1$ can permit synchronization of the network by $\beta_2$, an effect resembling the conditioning role of the HD pathway. This mechanism is supported by the finding that $\beta_2$ emergence was delayed by ~100ms following the suppression of $\beta_1$ activity (supplementary S5 Fig). This modulation of an auxiliary, non-targeted rhythm at $\beta_2$ is consistent with experimental results reported by Sanabria and colleagues, in which phase locked stimulation of the GPi led to higher frequency beta activity (>18 Hz) in tandem with targeted suppression of rhythms at 11–17 Hz [26]. This phenomenon is not limited to the CBGT circuit and has been observed in the emergence of secondary tremor rhythms following phase-locked thalamic DBS [27]. Our analysis of using different amplitude thresholds for gating stimulation (supplementary S4 Fig) suggests that frequency specificity is reduced at lower thresholds (such as those used in [26]). This may accentuate effects on non-targeted frequency bands (i.e. amplification of $\beta_2$). These findings emphasize the importance of optimizing the gating threshold.

To experimentally test our predictions that phase specific stimulation can induce changes in circuit-wide synchronization, it would be necessary to make extensive multi-site recordings during phase locked stimulation. Experimental setups for electrical stimulation have recently been described in 6-OHDA lesioned rats [31] which has the potential to test the PD specific aspects of this work, such as the conditioning role of HD and its potential to be mimicked by stimulation. State dependent effects could be also be tested by inducing pathway specific changes, either via pharmacological, optogenetic [51], or electrical [57] means, and then analysing the effects of phase specific stimulation across a circuit of interest. Recent work outside of the CBGT circuit, has shown that optogenetic stimulation can achieve phase specific modulations of network-wide activity similar to that described here [58].

## Informing strategies for future stimulation paradigms

Here we model cortical stimulation, locked to activity sensed in the STN, a configuration chosen as it provided better suppression of STN β1 rhythms than the reverse stimulation/sensing combination (supplementary S3 Fig). Stimulation was modelled as a sinusoidal input to the superficial pyramidal cells, similar to that resulting from transcranial alternating current stimulation (tACS) which has been shown to entrain sub-threshold cortical activity [59] in a phase dependent manner [60]. Thus, predictions from this theoretical model could be tested using a combined depth electrode implanted in the STN and closed-loop control of tACS stimulation. It should be noted that stimulation of the STN phase locked to cortical activity can also be used to achieve similar effects if stimulation amplitude and thresholds are optimised according to the cost function of interest (e.g. β1 suppression; supplementary S4 Fig).

Despite this, experimental evidence for therapeutic effects of cortical stimulation are inconsistent (see review [61]), yet no closed-loop approaches as presented here have been tested so far. The dual effects of phase specific stimulation (i.e., achieving both amplification and suppression), combined with recent evidence that cortical stimulation can suppress pathological activity in vivo [51,62] highlight the clinical potential for well-timed cortical stimulation. Nonetheless, our modelling suggested that although STN stimulation paired to cortical sensing

exhibited inferior state recovery (supplementary S7 Fig), it can suppress STN $\beta_1$ (S4 Fig). This setup could be readily implemented with current technology leveraging paired ECoG recordings with DBS [32,34].

Our model contains a number of important parameters, such as stimulation amplitude, frequency, shape etc., that are not explored here. These quantities form important targets for optimization, that can facilitate recovery of a range of network states. We used a high threshold for burst detection, and to gate stimulation. This resulted in a trade-off (supplementary S4 Fig) that allowed us to clearly illustrate phase dynamics, but in tandem, limited the efficacy of the stimulation. Furthermore, the spatial precision of stimulation delivery is likely to be a limitation in real world applications, thus understanding better the layer or nuclei specificity of effects remains a goal of future work.

We also note the dependency between phase specific stimulation and the signal quality at the sensing electrode. Our supplementary analyses (see S4 Appendix) show that the zero-crossing technique used for phase detection is only effective at suppressing beta frequency activity up to around -5dB (~1:3 SNR). Real world applications will likely require more refined phase estimation techniques [63], especially those making explicit estimation of observation noise [64].

Finally, the finding that stimulation effects are themselves dependent on network state agrees with previous studies exploring the state-dependency of stimulation [65,66]. These findings support hierarchical control algorithms that nest low-level stimulation delivery (e.g., online phase estimation) beneath a state estimation scheme to dynamically adjust stimulation parameters (e.g., stimulation phase). This would allow stimulation to adapt to factors such as pharmacological treatment and sleep, minimize adverse effects such as dyskinesias [32], and potentially respond to synaptic reorganization resulting from therapy [57].

## Limitations

Neural mass models allow the interactions between large populations of neurons to be described as a low dimensional system. This makes them amenable to parameter optimisation, yet the approximations made can render them unsuitable to investigate phenomena such as complex spiking and nonlinear integration of inputs. In this model, we were unable to capture the direct biophysical effects of inputs representing high frequency stimulation on lower frequency activity, as the lumping of time constants lead to an effective filtering of high frequency drive. Possible ways of modelling high frequency stimulation include either biasing the parameters of the sigmoid function representing STN outputs to favour tonic activation or bringing endogenous noise to a level such that external inputs can saturate the same sigmoid. Both mechanisms induce beta suppression in this model via switching STN output from bursting to tonic activity, as proposed in [67].

Moreover, rhythms at high beta/gamma rhythms are abstracted in our model by a bimodal spectrum that results from an inability to capture higher frequency activity thought to be important in prokinetic states [32,36]. The model also omits low frequency alpha/theta which has been implicated in this circuit in dystonia [68] but were not present in the experimental data used to fit the models. Furthermore, models were fit to activity recorded under urethane anaesthesia and may be only qualitatively similar to that seen in the awake animal. This may overemphasize pathways such as thalamocortical feedbacks that exhibit increased strength under anaesthesia [69]. However, urethane's effects on neurotransmitter function are small compared to other anaesthetics [70], and dose dependency allows access to cortically activated states which most resembles the waking brain [35].

Model extensions could incorporate additional neuronal populations, such as pallido-striatal arkypallidal cells [71], multiple cortical sources of hyperdirect pathway, distinct populations

of striatal medium spiny neurons, or pallidal interneurons. Specifically, inclusion of arkypalli-dal projections would also allow us to predict how alternative models considering pallido-stria-tal origins of beta oscillations [72] may alter response to stimulation. These additions to the circuit model would render a larger number of states recoverable by phase locked stimulation, but also demand increased spatial and temporal specificity to access them. Furthermore, the dataset here is limited in its ability to answer these questions due to the paucity of spatial reso-lution in the cortical recordings (ECoG from a single cortical "screw").

Finally, the model of stimulation presented here does not incorporate the spatial complexity of real life epicortical stimulation, rather we model the stimulation as current injection directly to a single neuronal population. Applications of this technology *in vivo* would require detailed understanding of the spatial extent of stimulation to understand how electrical spread could affect pathway specific activity.

## Conclusions

The outcomes of this work are twofold: (1) the expression of rhythmic biomarkers (e.g., STN beta bursts) are shaped by the network connectivity (e.g., connectivity of the CBGT circuit); and (2) precisely timed stimulation can recover network activity in a way that could compen-sate for disruption resulting from synaptic reorganization (e.g., alterations of the HD or PS pathways). By resynchronizing population activity to reflect physiological states, stimulation could restore neural communication without the need to alter connectivity. This model allows for numerous sensing and stimulation regimes to be explored that can assist the development of next generation stimulation that can flexibly pattern neural activity to restore healthy neural communication.

## Materials and methods

### Ethics statement

All experiments were conducted in accordance with the Animals (Scientific Procedures) Act, 1986 (United Kingdom), and with Society for Neuroscience Policies on the Use of Animals in Neuroscience Research.

### Electrophysiological recordings in 6-hydroxydopamine (6-OHDA) lesioned rats

Parameters of a computational model were constrained using a set of archival data consisting of multisite recordings in the basal ganglia and cerebral cortex of nine adult male Sprague-Dawley rats (Charles River, Margate, UK) with 6-OHDA induced dopamine depletion, a model of degeneration associated with Parkinsonism in humans described previously [73,74]. Animals were implanted with two multi-contact silicon probes to measure local field potentials (LFP) from multiple structures in the basal ganglia: dorsal striatum, external segment of palli-dum (GPe), and subthalamic nucleus (STN). Additionally, electrocorticography (ECoG) was measured over "secondary motor cortex" (M2), a homologue of the premotor cortex in humans [75], using a 1 mm diameter steel screw juxtaposed to the dura mater above the right frontal cortex. Anaesthesia was induced with 4% v/v isoflurane (Isoflo, Schering-Plough Ltd., Welwyn Garden City, UK) in O2 and maintained with urethane (1.3 g/kg, i.p.; ethyl carba-mate, Sigma, Poole, UK), and supplemental doses of ketamine (30 mg/kg; Ketaset, Willows Francis, Crawley, UK) and xylazine (3 mg/kg; Rompun, Bayer, Germany). Recordings were made during periods of 'cortical activation' [69] induced by a hind-paw pinch.

For more details of the experimental recordings and data acquisition please see the original experimental papers [15,35,50,74].

All data (LFP and ECoG) were: 1) down sampled from the hardware native 17.9 kHz to 250 Hz using Spike2 acquisition and analysis software (Cambridge Electronic Design Ltd., Cambridge, UK); 2) imported into MATLAB; 3) mean subtracted; 4) band-passed filtered 4–100 Hz with a finite impulse response, two-pass (zero-lag) filter with order optimized for data length; 5) Z-scored to standardize to unit variance; 6) divided into 1 second segments; and 7) subjected to a Z-score threshold criterion such that epochs containing any high amplitude artefacts were removed. The exact threshold was chosen on a case-by-case basis dependent upon recording gain and size of the artefact. Example traces of the recordings can be seen in Fig 1A. Artefact-rejected, epoched data were then taken forward to compute data features for the fitting procedure.

## Data features used for model estimation: Spectra and directed functional connectivity

Local dynamics within each node of the model were constrained by fitting to power spectra. Power spectra were constructed using Welch's periodogram method computed using non-overlapping epochs (1 second) multiplied by a Hanning window. To reduce spectra to their main peaks, the 1/f background of the empirical spectra was removed by first performing a linear regression in the log-log space and then subtracting the linear component from the spectra [76,77]. This ensured that the parameter estimation scheme was focused upon fitting the spectral peaks in the empirical data and not the background noise. The spectra from the nine rats were then combined using the mean across the group.

To constrain interactions between connected populations we used non-parametric directionality (NPD), using the Neurospec toolbox (http://www.neurospec.org/). NPD provides a non-parametric assessment of directed connections between neural signals derived from their spectral estimates alone [78,79]. Briefly, NPD performs a prewhitening of the signals' autospectra that allows for the (symmetric) coherence to be decomposed into its (asymmetric) directional components by integrating over separate lags of the (prewhitened) cross-correlations and transferring back to the frequency domain. The resulting NPD spectra were averaged across the nine rats.

The group averaged spectra and NPD were further smoothed using a sum of Gaussians (maximum of three), with order selected dependent upon the best fit evaluated using the adjusted $R^2$. These data features form the summary statistics upon which the model was inverted. The individual power spectra for each region are shown in Fig 1B (insets), and full set of features show in supplementary S1 Fig.

It should be noted that dynamics not summarised in these features such as properties of bursting activity, do not inform parameter estimates and thus cannot be expected to constrain model behaviour.

## Model description

We used a model describing the large-scale activity of coupled neuronal populations within the cortico-basal ganglia-thalamic network. This model has been used previously to recapitulate spectral features of recordings made in Parkinsonism [50,80,81]. We modified the model to explicitly incorporate stochastic inputs that could give rise to bursting dynamics (see Materials and Methods section "Formulation of Model from Coupled Neural Mass Equations"). The model includes a motor cortex microcircuit consisting of three pyramidal layers (superficial, middle, and deep) plus an inhibitory interneuron population [82]. Each cortical layer also

contains recurrent self-inhibitory connections reflecting local neuronal gain control. Furthermore, we modelled subcortical neuronal populations of the basal ganglia: the striatum (STR), the external and internal segments of the globus pallidus (GPe/i), the subthalamic nucleus (STN); as well as the ventrolateral thalamus (Thal.). Intrinsic noise arising from synaptic stochasticity, and indeterminate background activity was modelled as stochastic inputs to all the subcortical populations as well as the middle pyramidal layer of the motor cortex. The precise model architecture (depicted in Fig 1B) was selected using a model comparison procedure (described in Materials and Methods section "Parameter Estimation and Model Selection").

### Formulation of model from coupled neural mass equations

The circuit model comprises six regions/sources (i.e., the putative origins of empirically recorded field activity) that are themselves made up of a set of one or more locally coupled populations (e.g., supragranular cell layer in motor cortex). Each population in the model is described by a coupled neural mass equation [83,84] that models the average voltage change in a large, homogenous population of neurons. Overall, the model consists of nine coupled $2^{nd}$ order stochastic delay-differential equations (separable into 18, $1^{st}$ order equations given in S1 Appendix).

To model long distance connectivity between each source (e.g., M2 → STR), contributions from the $m^{th}$ to the $n^{th}$ source are delayed according to the delay matrix D. The connection strength is given by a weighted connectivity matrix $\omega$, and the total input to source $n$ (given in the superscript) at time $t$, is given by the sum of inputs across all 6 sources:

$$A^n(t) = \sum_{m=1}^{6} \omega_{n,m} s_n(V^m(t - D_{n,m})) \qquad (1)$$

where $V^m$ is the output voltage of the $M^{th}$ source, $D$ is the delay matrix specifying the delay for connection of source $m$ to $n$. Long distance connections are assumed to form substantial delays that are explicitly incorporated into the model (constraint such that $D_{n,m}>0$). The average output spike rate of the population in response to a voltage $v$ is given via the sigmoid operator:

$$S_i(v) = 1/(1 + e^{-R_i v}), \qquad (2)$$

which is parameterised by $R_i$ to determine the slope of the activation function (a parameter specific to each of the $i^{th}$ populations) and effectively models the variance of the population's firing thresholds.

The connectivity matrix of the full model (depicted in Fig 1B) is given below:

$$\omega = \begin{bmatrix} 0 & 0 & 0 & 0 & 0 & \omega_{1,6} \\ \omega_{2,1} & 0 & 0 & 0 & 0 & 0 \\ 0 & \omega_{3,2} & 0 & \omega_{3,4} & 0 & 0 \\ \omega_{4,1} & 0 & \omega_{4,3} & 0 & 0 & 0 \\ 0 & \omega_{5,2} & 0 & \omega_{5,4} & 0 & 0 \\ 0 & 0 & 0 & 0 & \omega_{6,5} & 0 \end{bmatrix} \qquad (3)$$

where column 1 gives connections projecting from M2; column 2 from the STR; column 3 from the GPe; column 4 from the STN; column 5 from the GPi; and column 6 from the Thal. Equivalently the rows give the weights of the input to each of the populations. Variants of this full model can then be created by adjusting the parameters or removing coefficients $\omega_{n,m}$ from the matrix. Equations were integrated numerically using the Euler-Maruyama scheme as detailed in S2 Appendix. Example traces of the model's simulations can be seen in Fig 1C.

## Parameter estimation and model selection

The parameters and architecture of the model were estimated using a scheme based on the sequential Monte Carlo Approximate Bayesian Computation algorithm (ABC; [85–88]). ABC is an algorithm for simulation-based inference [89] and allows for inversion of nonlinear, stochastic time series models. Importantly, this permits the investigation of bursting dynamics, such as those analysed in this work. The validity of this approach (in terms of the accuracy of estimation of parameters as well as the identification of model architectures), given the type of neural mass models and neurophysiological data described here, has been examined in previous work [90]. Briefly, ABC approximates the posterior density over models and their parameters, given some empirical data. This is achieved by computing the forward simulation using $N$ draws from a prior distribution of parameters and then iteratively rejecting samples dependent on the distance between the simulated and empirical data. Explicitly, experimental data and simulations are compared by first transforming the two datasets using a common summary statistic and then computing the goodness-of-fit, in terms of the mean squared error across the features. Here, we use the power spectral density and directed functional connectivity (see Materials and Methods section "Data Features used for Model Estimation: Spectra and Directed Functional Connectivity") to capture oscillatory dynamics and their interactions between neural populations, respectively. By adaptively reducing the threshold on the acceptable error between the summary statistics of the empirical and simulated data, the algorithm converges towards an approximation of the true posterior density over parameters, by traversing a sequence of intermediate distributions. Convergence was determined by setting a threshold on the error gradient (i.e., the improvement in accuracy with each step).

The model structure was determined by fitting 13 different models (described in S3 Appendix) to the data and then performing a model comparison using the estimate of the marginal probability as the criterion for the best fitting model. Marginal probabilities (i.e., the approximate model evidence) were estimated by drawing $N$ times from the posterior to simulate a set of $N$ realizations of the summary statistic. The marginal probability was then given by the probability that the summary statistics were less than a certain threshold $\epsilon^*$ (common across models) distance from the actual data (see S3 Appendix for more detail). The outcome of this procedure is given in [90] and in which we found that a model incorporating both the hyperdirect and subthalamo-pallidal pathways was the best candidate in describing the patterns of neuronal activity in recordings made in Parkinsonian rats. This model is very similar in architecture to previous work investigating the same system [50,81]. This posterior model fit was used for the simulations in this paper and is referred to as the *fitted* model. Its architecture is depicted in Fig 1. Specifically, we used the maximum a posteriori estimate (the mode of the marginal posterior distribution over parameters) of parameter expectations to specify the values of the fitted model. For a full list of model parameters that were estimated and details of their priors, please see table in S1 Appendix.

## Definition of discrete network states

For the purposes of this paper, we defined a set of discrete network states (i.e., configuration of connection weights) that could be used to explore the model's dynamics and its response to stimulation. As was introduced, dopaminergic cell loss linked to Parkinsonism has been associated with the weakening of the HD pathway [8–10], as well as strengthening of the PS pathway [11,12]. For each of these connections, we set both an *Up-* and *Down-*regulated state reflecting changes in connectivity from the model fit to 6-OHDA data. Previous experimental work highlighted a 70% increase in the amplitude of inhibitory post synaptic currents at the STN following dopamine depletion [11], equating to a *PS-Down* state with roughly 30%

connectivity of the 6-OHDA lesioned model (i.e. fitted model). We set a *PS-Up* state with connectivity strength that elicited roughly a doubling of broadband β power in the STN (c.f. upper range of effective beta band modulation reported in [25]). Similarly, we set a *HD-Down* state also at 30% connectivity of the fitted model, and an *HD-Up* state yielding a doubling of broadband β power. We constructed states reflecting both an increase and decrease in connectivity with the aim of later comparing them to the outcomes of phase specific stimulation (see Estimating the Recovery of Network States by Stimulation) which was expected to both amplify and supress rhythms [25].

Altogether, we defined two hyperdirect states: *HD-Down* and *HD-Up*; and two pallido-sub-thalamic states: *PS-Down* and *PS-Up*. In our discussion of the results, we consider *HD-Up* and *PS-Down* as proxies for the dopamine intact state in which motor symptoms are expected to be alleviated, whilst *HD-Down* and *PS-Up* represent further synaptic reorganization (i.e. beyond that of the 6-OHDA lesioned animals to which the model was fit) reflecting hypothetical disease progression and increased severity of motor symptoms. Evidence that motor deficits may be worsened further in the pathological state includes both optogenetic stimulation at beta frequencies of the 6-OHDA rat model [51], as well as 20 Hz DBS delivered in patients with PD [91,92]. Schematics of these states and their hypothesised functional significance are depicted in Fig 2A–2C.

## Characterizing network states with spectral fingerprints

For each network state, we simulated 128s of data and then formed summaries of the population activity of the circuit by constructing a set of spectral fingerprints [33] that we would later use to classify stimulation outcomes. These fingerprints consisted of: (A) the spectra of the population activity (as shown in Fig 2D, 2E, 2G and 2H) computed using Welch's periodogram method (with 1 second segments) for each source in the model. Spectra for each of the 6 sources were truncated to the range 2–48 Hz and then normalized to unit variance to prevent large changes in amplitude dominating the comparisons. Following this, all spectra were concatenated into a single vector. (B) The matrix of all pairwise phase locking values (PLV) between each source (computed within-bursts, see Definition of Transient Burst Events. . .; shown in Fig 3B; for details of its computation please see Materials and Methods section "Phase Synchronization: Connectivity matrices and Time Resolved Estimates"). Empirical states were constructed in the same way- computing the fingerprints of spectra and PLV individually for each of the nine animals. Note empirically derived spectra were corrected for 1/f background (as detailed in Materials and Methods section "Data Features used for Model Estimation: Spectra and Directed Functional Connectivity"). When comparing empirical states, only PLV pairs within the basal ganglia were used. This was because the poor SNR of the cortical ECoG led to small PLVs that were not well reflected in the model.

Note that spectral fingerprints were computed "within-burst" (see Materials and Methods section "Definition of Transient Burst Events and Statistics") since (A) we aimed to characterize network properties that were directly attributable to high SNR in the beta range. Preliminary analyses (not included) indicate that SNR impacts spatial selectivity of network synchronization. (B) We later compare the fingerprints derived from spontaneous activity and those induced by stimulation. As we gate stimulation according to beta bursts, this ensured that analyses were directly comparable between the two conditions.

## Definition of transient burst events and statistics

To define and characterize the properties of intermittencies in rhythms, we constructed a band filtered signal and then used a threshold on the envelope (method summarised in Fig 3A;

[17,93]). Specifically, we used a zero-phase, fourth order Butterworth filter for either lower ($\beta_1$: 14–21 Hz) or upper beta ($\beta_2$: 21–30 Hz) frequencies. We then constructed the analytic signal from the band-passed signal using the Hilbert transform. The instantaneous amplitude and phase are then given by the magnitude and angle of the complex signal. Bursts were defined as periods at which the envelope of STN beta activity exceeds the 75[th] percentile (c.f. [17,93]) for a duration longer than one cycle of the lower cut-off frequency of the filter [93,94].

We employ a burst definition based on the 75[th] percentile of the amplitude envelope. This is an important parameter which influences the efficacy of stimulation, as it sets a bound on the gating of stimulation. This choice is driven by the dependency between instantaneous phase and SNR [95,96]. Note that in practice, stimulation can be delivered at much lower thresholds (e.g., 15[th] percentile, see [26]) or with no threshold at all (see supplementary S4 Fig). Recent work has highlighted the limitations of using a threshold to detect bursts, especially where a common threshold definition is applied across data [97,98]. Here, we avoid such a comparison, and compute thresholds specific to the given network states.

We use the term "within-burst" consistently to denote data across the network within the time window that a STN beta burst was detected according to the above criteria. Changes in within-burst amplitude (Figs 3D and 3I; and 4D) were compared against randomly selected out-of-burst data (from the same network state and matched to the number and duration of the detected bursts) using a non-parametric cluster-based permutation test (see Materials and Methods section "Statistical Comparisons").

## Phase synchronization: Connectivity matrices and time resolved estimates

In order to measure the synchronization between population activity within the model, we summarised phase consistency between pairs of simulated signals using the phase locking value (PLV; [99]) over $N$ time points:

$$PLV = \frac{1}{N}\left|\sum_{n=1}^{N} e^{i\theta(n)}\right| \tag{4}$$

where $\theta(n)$ is the phase difference $\phi_1(n)-\phi_2(n)$. Instaneous phase was computed as described for within bursts, using the angle of the Hilbert transform of the bandpassed signals (see Materials and Methods section "Definition of Transient Burst Events"). To estimate changes in the spatial pattern of phase synchronization across the network, the difference in the within-burst PLV between each network state (i.e. *HD-Up/Down*, *PS-Up/Down*) and the fitted model was computed (Fig 3). We used a surrogate approach to determine the statistical significance of changes in PLV between states (Fig 3B only): we randomly selected out-of-burst data (from the respective state) with length matched to that of the real bursts, and then shuffled the signal pairs (i.e., burst with signal $A_1$ and $B_1$ became $A_1$ and $B_K$, with $K$ indicating the index of a randomly selected burst; pairs were truncated to the shortest burst). This maintained the power spectral features but removed any temporal correspondance between the segments. Using these out-of-burst data (selected from each state), we computed the difference in PLV (i.e. compared to the fitted state). This was done across 500 permutations with a threshold established at the 95[th] percentile of the resulting distribution to establish significance of the respective changes.

When using connectivity matrices as a fingerprint of each state (depicted in Fig 5C), we used the magnitude of the PLV (as in Eq 4) computed using the within-burst activity of each network state or stimulation model.

To investigate the time evolution of phase synchronization across bursts (Figs 3G and 3L; and 4F), we computed the PLV between cortex and STN, within a sliding window (200ms

duration, 95% overlap). Changes in the time-resolved phase and PLV (Figs 3E, 3G, 3J and 3L; and 4E and 4F) were compared against randomly selected, length matched, out-of-burst data using cluster statistics (see Materials and Methods section "Statistical Comparisons").

Finally, we computed the relative phase stability (RPS) of the STN/M2 phase coupling:

$$RPS(t) = \left| \frac{d(\phi_{M2}(t) - \phi_{M2}(t))}{dt} \right| \qquad (5)$$

where || indicates the absolute value, and $d/dt$ the derivative with respect to time. We approximated the derivative using the finite difference. We report RPS values as the average value it took across the burst duration (Fig 3F and 3K). This measure indicates the absolute rate of change in phase difference between M2 and STN, with an RPS = 0 indicating a constant phase aligment (i.e. zero-change), larger values indicate that the phase difference between STN and M2 fluctuated more over time.

## Modelling phase locked stimulation of motor cortex using activity in the subthalamic nucleus

To explore the impact of phase-locked stimulation on beta bursts (Figs 4 and 5), we first simulated a baseline dataset (128s) from the fitted model. We then re-simulated the model with an identical noise process but included a model of on-line, phase-locked stimulation of the motor cortex triggered from beta band activity sensed in the STN. To perform on-line estimation of phase, we adopted a zero-crossing analysis similar to the hardware approach reported in [27]. This algorithm was used to construct a sinusoidal stimulation pattern with phase updated every 25ms. To derive this input, a zero-phase, 4th order, Butterworth filter with passband at $\beta_1$ frequency was applied to the past three seconds of data up to the current update step. Data were zero-padded for one second on either end to reduce edge artefacts. The frequency response of the filter was inspected to ensure proper cut-off behaviour around the target band. To avoid unstable envelope estimates occurring close to the most current timestep due to the Gibbs effect [100], a 25ms delay was used to determine stimulation gating. The envelope was then constructed using the absolute value of the Hilbert transformed signal.

Stimulation was applied depending on if the following criteria were met: (A) the envelope of the sensing population exceeded the 75th percentile of that measured in the baseline (i.e., unstimulated data), and (B) no stimulation had occurred in the previous 500ms. Stimulation was then delivered for 500ms. Criterion B was introduced to ensure settling of dynamics back to their unstimulated state (as confirmed by visual inspection of traces) and to prevent runaway excitation. The phase of stimulation was estimated by extrapolating from the last positive zero-crossing of the bandpass filtered signal nearest to the current time step, assuming an 18 Hz (i.e., the centre of the pass band) sinusoidal rhythm. Despite this assumption of fixed frequency, instantaneous frequency is expected to be dynamic. Assuming an upper limit of the deviation in frequency of ±2 Hz, we can expect a maximum error of ±40° in phase alignment of the stimulation patterning. A validation of this method is given in S4 Appendix.

This estimate was then used to reconstruct the estimated phase of the sensing population $\phi_{sense}$ for the next 25ms cycle. The external input to the stimulated population in the model is then constructed as:

$$u_{stim}^{ex} = \begin{cases} A \sin(\phi_{sense} + \Delta\phi_{shift}), & crit = true \\ 0, & crit = false \end{cases} \qquad (6)$$

where $\Delta\phi_{shift}$ represents the target phase shift of the stimulating input with respect to the

sensing population; and $A$ is the amplitude of the stimulation which is set to 1/4 of the standard deviation of the intrinsic noise to the stimulating population ($u_i$ in supplementary equations S1.5 (in S1 Appendix)). Please see the discussion for a consideration of the role of amplitude in our results. The stimulus is only non-zero if the gating criteria *crit* are met. Summed intrinsic and extrinsic input to the neural mass (replacing $u_i$ with $U_{stim}$) then becomes:

$$U_{stim} = u_{stim}^{ex} + C_{stim}u_{stim}^{in} \tag{7}$$

Phase-locked stimulation was delivered to the superficial layer of the cortex only (c.f. non-invasive stimulation such as that performed in [59]). The phase shift, $\Delta\phi_{shift}$ was swept from -π to +π radians in 12 bins (i.e., a 30˚ resolution). The resulting power spectra for each stimulation regime was analysed and the within band power plotted against stimulation phase to yield band-limited amplitude response curves (ARCs) in either the lower or upper beta bands. ARCs were constructed from epochs of data during which stimulation was delivered. Epochs were matched according to stimulation duration when comparing ARCs and the unstimulated model.

Alternative stimulation strategies were also tested. Specifically, we used phase non-specific stimulation at 18 Hz, using the above gating criteria but with fixed instantaneous frequency (i.e., constant phase). As a second control, designed to identify whether spectral effects resulted from the phase tracking algorithm, we also "played back" stimulation patterns [101] but in a different model instance, where the actual phase was no longer matched to the model dynamics.

## Estimating the recovery of network states by stimulation

In a final set of analyses shown in Fig 5, we investigated the hypothesis that phase-locked stimulation of the motor cortex relative to STN can modulate network activity to reproduce states seen previously during changes to connectivity. To do this, we compared fingerprints of the predefined network states (see Materials and Methods section "Definition of Discrete Network States") with the outcomes of phase locked stimulation. To compare fingerprints between network states and phase stimulation we used a pooled $R^2$ measure [102]:

$$R_{pooled}^2 = 1 - \frac{1}{N_f}\sum_{n=1}^{N_f}\left(\frac{\sum_{i=1}^{N_n}\left(y_{n,i}^{state} - y_{n,i}^{stim}\right)^2}{\sum_{i=1}^{N_n}\left(y_{n,i}^{stim} - \overline{y_{n,i}^{stim}}\right)^2}\right) \tag{8}$$

where $N_f$ is the number of features (i.e., $N_f = 2$: one vector from concatenated spectra and one vector from the flattened connectivity matrices), $y_n$ the feature under consideration, $N_n$ the length of $y_n$ and $\overline{y_n}$ the mean of the $n^{th}$ data feature. $R_{pooled}^2$ varies between -∞ and 1, with 1 indicating a perfect fit, and negative coefficients indicating that the average fit is worse than that of a straight line going through the mean. Note that when comparing simulated with empirical states, we ignored features including GPi and Thal., as these were not available in the original data. Due to differences in the phase dependent recovery of empirical states, curves from both control and lesion conditions were realigned with respect to the phase attaining maximum recovery of the lesion state. When computing the pooled measure, we include the circular estimate within the sum of Eq (8).

In a secondary analysis we repeated this but modulated the strength of connectivity in the HD and PS pathways from 30% to their maximum defined in terms of 200% evoked beta. These results were then represented as heatmaps of the $R_{pooled}^2$ for connectivity strength versus stimulation phase.

## Statistical comparisons

Summary statistics of phase angles were computed with the circular mean and standard deviation, whilst tests of differences in means were computed using the Watson-Williams test (in text descriptions and Fig 3C and 3H). Tests for differences in continuous data (Figs 3D–3G, 3I–3L and 4C–4F) were performed using non-parametric cluster-based permutation tests using an implementation based in Fieldtrip [103], using the t-statistic when comparing means. Clusters were detected from 500 permutations and a P = 0.05 two-sided alpha level. For clarity of presentation, clusters were restricted to the top five largest effects, and clusters smaller than 20ms in duration were excluded.

## Supporting information

**S1 Fig. Full set of empirical data features and model fits.**
(DOCX)

**S2 Fig. Analysis of multi-Gaussian fits to empirical data.**
(DOCX)

**S3 Fig. Alternative stimulation policies.**
(DOCX)

**S4 Fig. Analysis of stimulation effects when using different control parameters.**
(DOCX)

**S5 Fig. Additional analyses of phase locked stimulation effects.**
(DOCX)

**S6 Fig. Stimulation recovery of empirically derived network states.**
(DOCX)

**S7 Fig. Analysis of state recovery when using STN stimulation phase locked to activity sensed in the motor cortex.**
(DOCX)

**S1 Appendix. Extended Model Description and Table of Parameters.**
(DOCX)

**S2 Appendix. Integration of Stochastic Differential Equations.**
(DOCX)

**S3 Appendix. Candidate Model Space and Model Selection.**
(DOCX)

**S4 Appendix. Validation of the Zero-Crossing Procedure for On-Line Phase Estimation and Predicted Effects of Signal-to Noise Ratio.**
(DOCX)

**S1 Table. Table of External Software.**
(DOCX)

## Acknowledgments

We would like to thank Prof. Peter Brown, Dr. Ashwini Oswal, and Dr. Carolina Reis for their helpful comments on drafts of this manuscript. We thank Dr. N. Mallet for acquiring some of the primary data sets. This work uses a number of toolboxes, generously developed,

maintained, and shared by their respective authors (details in supplementary S1 Table) to whom we are grateful.

## Author Contributions

**Conceptualization:** Timothy O. West, Simon F. Farmer, Hayriye Cagnan.

**Data curation:** Peter J. Magill, Andrew Sharott.

**Formal analysis:** Timothy O. West, Hayriye Cagnan.

**Funding acquisition:** Peter J. Magill, Andrew Sharott, Vladimir Litvak, Simon F. Farmer.

**Investigation:** Timothy O. West, Peter J. Magill, Simon F. Farmer.

**Methodology:** Timothy O. West, Hayriye Cagnan.

**Resources:** Andrew Sharott, Vladimir Litvak, Simon F. Farmer, Hayriye Cagnan.

**Software:** Timothy O. West.

**Supervision:** Vladimir Litvak, Simon F. Farmer, Hayriye Cagnan.

**Visualization:** Timothy O. West.

**Writing – original draft:** Timothy O. West, Simon F. Farmer, Hayriye Cagnan.

**Writing – review & editing:** Timothy O. West, Peter J. Magill, Andrew Sharott, Vladimir Litvak, Simon F. Farmer, Hayriye Cagnan.

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
