## [Decision Letter · Decision Letter 0]

30 Jul 2021

Dear Dr. West,

Thank you very much for submitting your manuscript "Stimulating at the Right Time to Recover Network States in a Model of the Cortico-Basal Ganglia-Thalamic Circuit" for consideration at PLOS Computational Biology.

As with all papers reviewed by the journal, your manuscript was reviewed by members of the editorial board and by several independent reviewers. In light of the reviews (below this email), we would like to invite the resubmission of a significantly-revised version that takes into account the reviewers' comments.

The reviewers were unanimous in their opinion that major issues with this manuscript need to be addressed.  These include the match and relevance of results to experimental data, the justification of the computational approach, the clarity of the methods section, the clarity of the overall message of the paper, and the quality of the discussion including discussion of limitations of the work.

We cannot make any decision about publication until we have seen the revised manuscript and your response to the reviewers' comments, including both the general and specific comments of reviewer #3. Your revised manuscript is also likely to be sent to reviewers for further evaluation.

Sincerely,

Jonathan Rubin

Associate Editor

PLOS Computational Biology

Daniele Marinazzo

Deputy Editor

PLOS Computational Biology

Reviewer's Responses to Questions

**Comments to the Authors:**

Reviewer #1: This manuscript describes the implementation and analysis of a model of the cortical-basal ganglia loop relevant to the contributions of abnormal oscillatory neural activity present in PD and suppression of this activity (and symptoms) by brain stimulation. The results are potentially interesting, but the initial paramterization of the model yields only a modest match the (quite) limited experimental data, the focus on beta “burst” activity is not well justified, especially in light of prior results of phase-specific stimulation in vivo (Sanabria et al), and the analysis of the therapeutic potential of phase-specific stimulation must be compared to other “control” forms of stimulation to put the effects in context.

Major:

The analysis of the ability of phase specific stimulation to restore “healthy” brain states is unclear and the presentation of these results is somewhat convoluted. Indeed, the assertion, “the ability to accurately recover states related to a deprecated HD pathway, suggests that M2 stimulation phase locked to STN may have the potential to restore network activity related to this pathway following degeneration related to dopamine depletion.” is somewhat upside down, as presumanly one would want to use stimualtion to replicate the HDup state (and similarly upside down for the PSup state). Prior to an extensive analysis of a range of PS and HD connection strengths (many of which seem extreme), it would be much clearer to create a prototype of a PD state, with contributions from both PSup and HDdn, and examine effects of phase-specific stimulation therein, and specifically the ability to recover the fully connected state. It is not clear clear in the analyis of 5G-L what the match is to the fully connected (ie, healthy) network, and it is not therapeutically relevant to access all network states in the unhelathy network.

The assertion that, “the potential for precisely timed stimulation to compensate for weakening of corticosubthalamic transmission” and “may be restored by well-timed stimulation” require comparisons to phase independent stimulation, as continuous stimulation can rescue the PSup and HDdn cases, as observed exprimentally. Further, the effects of phase specific stimulation appear suprisingly modest. Although it is stated that, “Maximal suppression of β1 is close to that reported in experimental work”, this is clearly not the case, where in PD, the beta power can be dramatically suppressed by either levodopa or continuous DBS. And similary, stimulation herein has quite modest effects on beta amplitude within bursts (4D).

The focus on and relevance of beta bursts in the present context is not clear, and this raises several imporant issues. First, there are substantial limitations to using a threshold to detect burst activity (see for example Anderson et al 2020 and Schmidt et al 2020) and average burst activity cannot be separated from average beta amplitude. Second, there are increases in bursts for HDup as well as HDdn, and the increase in beta amplitude at the “onset of burst” (3D and 3I) is a circular argument as this is how the bursts were defined to begin with. Third, the exclusive focus is on phase alignment within bursts, but also imperative to consider phase alignment otherwise, and the within burst phase in Hdup is not considered (and does not fit with broad assertions). Fourth, and with respect to the importance of bursts vis a vis phase specific stimulation, this is overstated in the Introduction, as Sanabria et al observed similar results using phase-locked stimulation in vivo without reliance on burst activity. Further, as Sanabria et al have demonstrated the utility of phase-locked stimulation, the reader should be reminded that this is not the first demonstration of phase-locked stimulation reducing beta amplitude in the Discussion.

The model appears to generate a narrow beta band oscillation, but overall a somewhat marginal match between the experimental spectra an the model spectra, even after extesnive avergaging and smoothing of the former. (Note, the line thickness and large spaces between dashes in the dashed lines makes it very difficult to compare the spectra in S1).

Several important limitations are not addressed in the Discussion. First, the experimental data used to compare the model were collected under a complex cocktail of anesthetics and this can influence brain oscillatory activity. Second, the mtor cortical stimulation is delivered as an intracellular sine wave, and this has substantially different effects than either epicortical or intracortical (extracellular) stimulation.

Minor:

It is not readily apparent that cDBS targets factor B. The authors suggest reduction in beta oscillations is sufficient for this claim but the same claim could be made for levodopa.

Not sure that (presumably multi-synaptic) functional connectivities’ "spectral fingerprints" should be equated to known pathways such as the hyperdirect pathway (monosynapic connections).

The radial dimension in Figs 3C&H 4C is not clear. In Fig 4, suppressive stimulation appears to decrease every reported metric except this (unlabeled) one. At different radial lengths, the same SD will appear to take up more space, so variability on this axis should have some justification.

Many times the authors refer to "transient phase synchronization" but show something a little bit different, the phase synchronization during transient amplitude events. True analysis of "transient phase synchronization" would be agnostic to amplitude values.

The changes in phase near the onset of the bursts (Fig 3E,J) are difficult to interpret, as there are opposite changes in realtive phase across pro- vs anti-PD changes in connection strength, as well as substantial changes in phase at times other than near burst onset.

The correlations of phase stability and beta amplitude (3F,K) are just that – correlations, and thee obserations, contradicted inone of four cases, are not supportive of the statement of causation, “bursts persist for the duration of a stable period of phase locking between STN and M2”.

The tenor of the statement, “high amplitude beta oscillations that are known to be electrophysiological correlates of symptom severity”, is too strong as there are myriad exceptions to this relationship, and it is most strong only for bradykinesia.

The error that was used to compare iteratively versions of the model to the experimenal data during the fitting process is not clearly described and more specific information beyond, “the distance between the simulated and empirical data”, is necessary.

The motor cortex is asserted as an ideal stimulaton target, yet clinical stimulation of motor cortex in PD has led overall to highly vaiable and mostly disappointing results.

The assertion, “this agrees with studies implicating increased coupling of the STN/GPe loop in the emergence of excess beta synchrony in the CBGT circuit” should consider other (competing) hypotheses on the origins of increased bet synchrony including striatial and cortical sites.

Not clear that R2 is an adequate metric to assess similarity of concatenated spectra, as values are largely bimodal.

Fig 4B insets too small to be legible.

Figure 4 legend title inverted stimulation and recording locations and is incorrect.

“Data” is plural and verbs should be congugated as such.

Line 552 “that are themselves be made up of”, delete “be”

Line 427 “Our results show that the expression STN beta” needs an “of”

Avoid jargon “spectral fingerprints”, “phase slip”, ‘open a channel for synchronization", …

Reviewer #2: Thank you for inviting me to review this manuscript. My main expertise is human and clinical neurophysiology, so my review mainly focuses on physiological plausibility of the findings and the clinical outlook. Overall, this is a very valuable and innovative approach to computational network models of the basal ganglia. It innovates previous approach in multiple domains, as it reliably produces oscillatory population activity and connectivity, allows characterization of temporal dynamics that have recently been the focus of DBS neurophysiology research and investigates the effect of phase specific closed-loop stimulation that is currently on the rise for neurotherapeutics. The main finding that phase specific stimulation has the potential to compensate for pathological network disruption is impactful.

I have few comments that should be addressed:

1. One of the major results of the model dynamics is the definition of a shift from a healthy state associated with hyperdirect pathway driven activity with low pallidosubthalamic connection strengths to a severely dopamine depleted state with lower hyperdirect drive and increased coupling in the pallidosubthalamic pathway. Similar patterns have indeed been reported in experimental animal studies (Tachibana 2011). However, the latter and other studies have also demonstrated that firing rates in the parkinsonian state in the STN are significantly increased. How can the authors explain this reproducibly demonstrated increase in firing of the STN if the major glutamatergic input is decreased and the pallidosubthalamic inhibitory input is increased? Can this be contextualized to the effect of dopamine on pathway specific striatal spiny neurons?

2. The nature of simulated stimulation should be clearly stated in the results, to guide the reader regarding the comparability to available treatment strategies. Even after diving into the methods, it was impossible for this reviewer to understand whether DBSlike high-frequency stimulation, single pulse stimulation or oscillatory “currents” or a non-electrical network proxy of stimulation was applied. Would the stimulation be testable in animal models or is it abstract? To increase the translational value of the submission, I would suggest to elaborate on the available neuroscience methods that could experimentally reproduce the results in animals or humans.

3. In a similar vein, given the relevance of the paper to the DBS community the localization of sensing and stimulation for future applications should be stated more clearly. To this reviewer it was hard to understand why cortical instead of subthalamic phase specific stimulation was implemented. Most of the times, stimulation is mentioned without stating stimulation location, which could be confusing to clinicians who expect subthalamic stimulation that is readily testable in patients. This is aggravated by the introduction that motivates the results section with phase specific DBS. Now cortical stimulation would not be considered deep brain stimulation, but would the authors suggest electrocorticography as a potentially relevant sensing and stimulation approach? This should be discussed.

4. The authors state in the first paragraph of the results that a qualitative feature of significant feedback of beta oscillations from subcortex to cortex could be produced by the model. To the best of my knowledge, all directionality analyses in human patients have demonstrated that beta activity is driven by cortex (e.g. Litvak et al., 2011).

5. The manuscript uses a lot of non-standard abbreviations, making it very hard to read and follow. I would suggest to try and cut down on the number of abbreviations and placeholder terms to improve comprehensibility.

Reviewer #3: General comments

This study examines changes in network connectivity and the effect of phase-dependent stimulation in a neural mass model of the cortico-basal ganglia-thalamic circuit. It is concluded that the pattern of beta synchrony in the network depends on connectivity and that well-timed stimulation of the cortex can recover the original state of the network.

The questions addressed are interesting and topical. However, the message is not very clear. The results are quite long and very discursive, making them difficult to follow or to identify the main findings. In comparison, the Discussion is quite ‘light’ and doesn’t address the nuances of the model or the details of mechanisms involved. A thorough comparison with previous studies is missing and the limitations of the study and model are not addressed.

There is a question around the bimodal distribution of the power spectrum of the simulated neural field data. This bimodal distribution does not appear to be present in the experimental data on which the model is based and constrained. It is not clear what the mechanism underpinning this bimodal distribution in the model is or whether similar effects occur physiologically. When introduced in the results the bimodal distribution is linked to other experimentally reported data, but how does this fit with the data on which the model has been based, and why are there differences between the two? Also, why did the optimization of model parameters converge on this model which appears different to the experimental data, does the model still represent the experimental data/states?

The type of model used, i.e. a neural mass model, is not described until page 2 of the Methods. This should be clarified in the Abstract, Introduction and Results. Similarly it should be clarified here that the spectral ‘fingerprints’ referred to are at the population level.

Specific comments

What is the motivation for targeting the cortex for stimulation? This should be outlined in the Introduction.

GPe-GPe interconnectivity is not considered in the model. How would this impact the results?

The model describes well the power spectrum of the population of neurons and directed functional connectivity within the network. However, is this is a true representation of the network state as the behavior of individual neurons is not captured?

Figure 1 and 2 indicate connections from the secondary motor cortex (M2) only to STN. However, the STN receives projections from the primary motor cortex (MI), the supplementary motor area (SMA), and premotor cortex (e.g. Nambu et al.). Is this a reasonable assumption in the model?

Down regulation of the hyperdirect pathway is an assumed condition of the Parkinsonian/dopamine depleted state (p. 24) based on references [9-11]. How does this fit with the conclusions of increased effective connectivity in the hyperdirect pathway obtained using similar Bayesian modelling methods in the same data set in the model referred to in [66]?

What happens when the HD pathway strength is decreased and PS strength is simultaneously increased? Why were changes in the hyperdirect and pallido-subthalamic connection strengths examined independently, rather than allowing simultaneous changes in both? This would seem to be a better representation of dopamine depletion/ worsening symptoms then changing them independently.

p. 11. Why was a sinusoidal stimulation chosen for the cortical stimulation? What was the frequency of this stimulation? This is obviously different to the type of stimulation that is delivered clinically.

The maximum suppression of beta power provided by the phase locked stimulation seems quite modest (at -30 % in the STN and -17 % in M2). In contrast, the phase locked stimulation was much more effective in increasing beta band power (up to 155 % and 240 % of the baseline value). What is the reason for this ‘asymmetric’ response?

**Have the authors made all data and (if applicable) computational code underlying the findings in their manuscript fully available?**

Reviewer #1: Yes

Reviewer #2: Yes

Reviewer #3: Yes

PLOS authors have the option to publish the peer review history of their article (what does this mean?). If published, this will include your full peer review and any attached files.

Reviewer #1: No

Reviewer #2: **Yes: **Wolf-Julian Neumann

Reviewer #3: No
---

## [Decision Letter · Decision Letter 1]

16 Nov 2021

Dear Dr. West,

Thank you very much for submitting your manuscript "Stimulating at the Right Time to Recover Network States in a Model of the Cortico-Basal Ganglia-Thalamic Circuit" for consideration at PLOS Computational Biology. As with all papers reviewed by the journal, your manuscript was reviewed by members of the editorial board and by several independent reviewers. The reviewers appreciated the attention to an important topic. Based on the reviews, we may accept this manuscript for publication, but this will depend on your ability to address the remaining issues raised by Reviewer #1 and to modify the manuscript according to the review recommendations.

Sincerely,

Jonathan Rubin

Associate Editor

PLOS Computational Biology

Daniele Marinazzo

Deputy Editor

PLOS Computational Biology

[LINK]

Reviewer's Responses to Questions

**Comments to the Authors:**

Reviewer #1: The authors are to be commended for the clear and comprehensive response to the prior review, and their extensive revision of what is now a much more clear and compelling manuscript. Nonetheless several important points remain to be addressed.

An important remaining limitation is the absence of comparisons to clinical standard high frequency (130 Hz) STN DBS. The comparisons to other (quite low frequency modalities) is a step in the right direction, but, given the added complexity of implementing the proposed phase-specific stimulation, a comparison to the remarkably simple open loop 130 Hz is required.

Further, several elements of the comparison of stimulation modalities (S3) are puzzling and require some further explanation. It is unclear in S3 why the power in the 0% condition with the phase specific stimulation is approximately twice as large as the baseline power in the other conditions that were tested? And not clear how this power spectra compares to those in Figure 2 or the changes in beta power in Figure 4B, where the increases in beta power at 330 degrees are much more than +42%, and there certainly does not look to be a ~50% increase in beta2 power?

The method relies on estimation of the phase of noisy signals and some quantitative evaluation of the effects of errors in phase estimation, as will occur with low SNR signals, in vivo, especially, when stimulation is effective at reduction of beta1 activity in STN. This will help to set specifications for implementation of the proposed approach.

The Limitations in the Discussion should consider those associated with the direct “intracellular” stimulation of a select subset of modeled cortical neurons and how this mifght compare to electric epicortical stimulation.

The variance accounted for metric in Fig 6A is puzzling as it is not clear how R^2 can either take on negative values or have absolute values greater than 1.

Color coding of spectra in bottom panel of Fig 5B looks swapped as power in STN with max B1 amplifying (blue) is greater than power with max B1 suppressive (orange)

Sanabria et al. stimulated GPi not STN.

Reviewer #2: I would like to thank the authors for their thorough discussion. All my comments have been addressed and I believe the manuscript has gained significantly from the revision.

Reviewer #3: The authors have completed a comprehensive revision of the manuscript. All of my comments have been satisfactorily addressed.

**Have the authors made all data and (if applicable) computational code underlying the findings in their manuscript fully available?**

Reviewer #1: None

Reviewer #2: Yes

Reviewer #3: Yes

PLOS authors have the option to publish the peer review history of their article (what does this mean?). If published, this will include your full peer review and any attached files.

Reviewer #1: No

Reviewer #2: **Yes: **Wolf-Julian Neumann

Reviewer #3: No

Figure Files:

Data Requirements:

Reproducibility:

References:

---

## [Decision Letter · Decision Letter 2]

31 Jan 2022

Hi Timothy -

As you will see below, your revisions have satisfied the reviewers (and me).  Congratulations on your impressive work.  I do have one remaining comment.  I do not want to delay the process any more, so I am happy to accept the paper and to leave this to your discretion.  The standard PLoS CB text below will tell you that you cannot make any changes to your manuscript from this point onwards other than a few specific types.  If you decide to make a minor edit to address my comment (e.g., in the page proofs review), that will be fine.  It may or may not come to my attention if that happens; if so, I will simply accept the change and move things along promptly.  Here is my comment:

%%%%%%%%%%%%%%%%%%%%%%

The revised text on L502-510 includes the statement:

"Whilst in this model it is not possible to directly simulate high frequency stimulation, it is possible to either enact a constant voltage change [62] or bias the sigmoid transfer function to favour tonic activation, a mechanism proposed by the model of Rubin and Terman [66]. Despite these limits in directly modelling high frequency stimulation, our model does however capture experimental features of phase specific stimulation at beta frequencies such as the emergence of secondary rhythms [26] and phase dependent modulation of cortical rhythms [30].”

It doesn't seem that this text is fully accurate. First, the response to reviews did include a simulation of high frequency stimulation. It seems that a more accurate statement is that in this modeling framework, stimulation imposed at high frequencies cannot impact power at lower frequencies, and thus the model cannot directly capture the biophysical effects of high frequency stimulation. Second, the response to reviewers showed that the 2nd approach based on the transfer function actually does attenuate beta in this model. Is seems like that could be worth mentioning. For these two reasons, a minor update of this text seems desirable.

%%%%%%%%%%%%%%%%%%%%%%%%%%

best regards,

Jon

Dear Dr. West,

We are pleased to inform you that your manuscript 'Stimulating at the Right Time to Recover Network States in a Model of the Cortico-Basal Ganglia-Thalamic Circuit' has been provisionally accepted for publication in PLOS Computational Biology.

Best regards,

Jonathan Rubin

Associate Editor

PLOS Computational Biology

Daniele Marinazzo

Deputy Editor

PLOS Computational Biology

Reviewer's Responses to Questions

**Comments to the Authors:**

Reviewer #1: The authors have responded to the comments.

Congratulations on an interesting and important contribution!

**Have the authors made all data and (if applicable) computational code underlying the findings in their manuscript fully available?**

Reviewer #1: Yes

PLOS authors have the option to publish the peer review history of their article (what does this mean?). If published, this will include your full peer review and any attached files.

Reviewer #1: No

---

## [Editor Report · Acceptance letter]

18 Feb 2022

PCOMPBIOL-D-21-01076R2 

Stimulating at the Right Time to Recover Network States in a Model of the Cortico-Basal Ganglia-Thalamic Circuit

Dear Dr West,

I am pleased to inform you that your manuscript has been formally accepted for publication in PLOS Computational Biology. Your manuscript is now with our production department and you will be notified of the publication date in due course.

With kind regards,

Livia Horvath
